# `Kona`: An Efficient Privacy-Preservation Framework for KNN Classification by Communication Optimization

Guopeng Lin [1]  Ruisheng Zhou [1]  Shuyu Chen [1]  Weili Han [1]  Jin Tan [2]  Wenjing Fang [2]  Lei Wang [2]  Tao Wei [2]

## Abstract

K-nearest neighbors (KNN) classification plays a significant role in various applications due to its interpretability. The accuracy of KNN classification relies heavily on large amounts of high-quality data, which are often distributed among different parties and contain sensitive information. Dozens of privacy-preserving frameworks have been proposed for performing KNN classification with data from different parties while preserving data privacy. However, existing privacy-preserving frameworks for KNN classification demonstrate communication inefficiency in the online phase due to two main issues: (1) They suffer from huge communication size for secure Euclidean square distance computations. (2) They require numerous communication rounds to select the $k$ nearest neighbors. In this paper, we present `Kona`, an efficient privacy-preserving framework for KNN classification. We resolve the above communication issues by (1) designing novel Euclidean triples, which eliminate the online communication for secure Euclidean square distance computations, (2) proposing a divide-and-conquer bubble protocol, which significantly reduces communication rounds for selecting the $k$ nearest neighbors. Experimental results on eight real-world datasets demonstrate that `Kona` significantly outperforms the state-of-the-art framework by $1.1\times \sim 3121.2\times$ in communication size, $16.7\times \sim 5783.2\times$ in communication rounds, and $1.1\times \sim 232.6\times$ in runtime.

## 1. Introduction

As a fundamental machine-learning technique, K-nearest neighbors (KNN) classification (Mucherino et al., 2009; Kataria & Singh, 2013) has drawn significant attention for its interpretability. The intuitive nature of KNN classification, which makes predictions based on the similarity of samples, allows users to easily understand and trust its outcomes, fostering its widespread adoption across diverse domains. For instance, in medical diagnosis, KNN can assist in identifying diseases based on patient data (Ali et al., 2020; Wang et al., 2020), while in recommendation systems, it can tailor suggestions to user preferences (Adeniyi et al., 2016; Singh et al., 2020).

The accuracy of KNN classification heavily relies on large amounts of high-quality data. However, in practical scenarios, high-quality data are usually distributed across different parties, and cannot be directly aggregated due to the sensitive nature of the data and regulatory restrictions. For instance, patient records are usually distributed across different hospitals or clinics and cannot be directly aggregated due to strict privacy regulations (e.g. GDPR (Voigt & Von dem Bussche, 2017)). As a result, collecting sufficient high-quality data to obtain accurate KNN classification faces significant challenges.

To address the need for aggregating sufficient high-quality data to perform accurate KNN classification, numerous privacy-preserving frameworks (Sun & Yang, 2020; Li et al., 2023; Wu et al., 2019; Wong et al., 2009; Cui et al., 2020; Liu et al., 2019) for KNN classification have been proposed. These frameworks typically employ multi-party computation (MPC) techniques, such as secret sharing or homomorphic encryption, to keep the sensitive data private throughout KNN classification. Therefore, these frameworks can utilize data from different parties to improve the accuracy of KNN classification while preserving data privacy, which broadens the applicability of KNN in real-world scenarios.

Despite the potential of these privacy-preserving frameworks for KNN classification, they still suffer from significant online communication inefficiency, which impedes their practical deployment. The online communication inefficiency arises from two primary issues: (1) Existing frameworks suffer from huge communication size for secure Euclidean square distance computations. Euclidean square distance is a commonly used metric to measure the similarity between samples in KNN classification. For a

---

[1]Fudan Univeristy, Shanghai, China [2]Ant Group, Shanghai, China. Correspondence to: Weili Han <wlhan@fudan.edu.cn>.

*Proceedings of the 42nd International Conference on Machine Learning*, Vancouver, Canada. PMLR 267, 2025. Copyright 2025 by the author(s).

dataset containing $n$ samples, performing a KNN classification requires computing the Euclidean square distance $n$ times. Existing frameworks, such as (Li et al., 2023; Liu et al., 2019; Sun & Yang, 2020), necessitate communicating secret shares or encrypted elements for each distance computation, which makes these frameworks inefficient, especially for large-scale datasets. (2) Existing frameworks suffer from numerous communication rounds for $k$ nearest neighbors selection. Selecting the $k$ nearest neighbors is a critical step during KNN classification. Existing frameworks, such as (Li et al., 2023; Liu et al., 2019), employ sequential bubble protocols to select the $k$ nearest neighbors, which requires $O(kn)$ communication rounds. The numerous communication rounds make these frameworks extremely inefficient, especially in wide-area network environments.

*Table 1.* Online communication complexity comparison between Kona and the state-of-the-art framework SecKNN (Li et al., 2023). Here, $n$ is the number of samples, $m$ is the number of attributes, and $k$ is the number of neighbors used in the KNN classification.

|  |  | Kona | SecKNN |
|---|---|---|---|
| Euclidean Square | Size | 0 | $O(nm)$ |
| Distance Computation | Rounds | 0 | $O(1)$ |
| K-nearest | Size | $O(kn)$ | $O(kn)$ |
| Neighbor Selection | Rounds | $O(k \log n)$ | $O(kn)$ |
| Total | Size | $O(kn)$ | $O(nm + kn)$ |
|  | Rounds | $O(k \log n)$ | $O(kn)$ |

In this paper, we present Kona, an efficient privacy-preserving framework for KNN classification. We resolve the aforementioned communication issues through two optimizations: (1) We propose novel Euclidean triples that enable secure Euclidean square distance computations without any online communication. We observe that although existing frameworks leverage random vectors (e.g. random pairs (Li et al., 2023)) generated in the offline phase to reduce the online communication size, these vectors are not fully compatible with Euclidean square distance computations. Inspired by the input-independent but function-dependent technique (Ben-Efraim et al., 2019), we tailor our Euclidean triples specifically for Euclidean square distance computations. This design eliminates online communication and does not introduce additional offline communication overhead compared to the state-of-the-art framework (Li et al., 2023). (2) We present a divide-and-conquer bubble protocol to significantly reduce the communication rounds required to select the $k$ nearest neighbors. We observe that the sequential bubble protocol incurs numerous communication rounds because secure comparisons are performed one by one. However, secure comparisons are largely independent of each other. Thus, our divide-and-conquer bubble protocol performs most secure comparisons in parallel, reducing the communication rounds from $O(kn)$ to

$O(k \log n)$. These two optimizations significantly enhance the efficiency of Kona, making Kona a promising solution for large-scale and real-world applications.

We summarize the main contributions in Kona as follows:

- We design novel Euclidean triples that enable secure Euclidean square distance computations without any online communication.

- We present a divide-and-conquer bubble protocol to significantly reduce the communication rounds required for selecting the $k$ nearest neighbors.

As is shown in Table 1, our proposed framework Kona significantly reduces both online communication size and communication rounds compared to the state-of-the-art framework, SecKNN (Li et al., 2023). Specifically, in terms of Euclidean square distance computation, Kona does not require any online communication, while SecKNN requires $O(nm)$ communication size. In terms of $k$-nearest neighbor selection, Kona requires only $O(k \log n)$ communication rounds, while SecKNN requires $O(kn)$ communication rounds. In terms of total communication overhead for KNN classification, Kona achieves reductions in both communication size and rounds compared to SecKNN.

To further demonstrate the efficiency of Kona [1], we compare the performance of Kona against SecKNN with eight real-world datasets, which encompass various domains and data characteristics. The experimental results demonstrate that Kona significantly outperforms SecKNN by $1.1\times \sim 3121.2\times$ in communication size, $16.7\times \sim 5783.2\times$ in communication rounds, and $1.1\times \sim 232.6\times$ in runtime. These results show that Kona is much more practical and scalable for real-world privacy-preserving KNN classification tasks.

## 2. Preliminaries

### 2.1. K-Nearest Neighbors Classification

KNN classification is a non-parametric, instance-based learning method that infers the class of a new sample by examining its similarity to known samples. Specifically, as is shown in Algorithm 1, KNN classification inputs a dataset of $n$ samples, each with $m$ attributes and an associated class label, and a new sample $\vec{a}$ to be classified, an integer $k$ representing the number of neighbors. To classify $\vec{a}$, KNN classification first computes the distance between $\vec{a}$ and every sample $\vec{x}_i$ in the dataset (Line 1-5). It then selects the $k$ samples closest to $\vec{a}$, known as its $k$ nearest neighbors (Line 6). Finally, it assigns the class label to $\vec{a}$ based on the majority vote among these $k$ neighbors (Line 7). This

---

[1] We have open-sourced the implementation of Kona at https://github.com/FudanMPL/Garnet/tree/kona.

straightforward approach makes KNN both conceptually simple and highly interpretable.

In this paper, we measure the similarity between samples by employing the Euclidean square distance, which is widely used in previous studies (Sun & Yang, 2020; Li et al., 2023; Liu et al., 2019) and is an intuitive metric that is easy to implement and interpret.

---

**Algorithm 1** $KNN\text{-}classify(\{\vec{x}_i, y_i\}_{i=0}^{n-1}, \vec{a}, k)$

---

**Input:** A dataset of $n$ labeled samples $\{\vec{x}_i, y_i\}_{i=0}^{n-1}$, where $\vec{x}_i$ is an $m$-dimensional attribute vector and $y_i$ is its class label; a new attribute vector $\vec{a}$ to be classified; and an integer $k$ representing the number of neighbors.
**Output:** A predicted label $b$ for $\vec{a}$.
 1: Initialize an empty array *dis*.
 2: **for** $i = 0$ to $n - 1$ **do**
 3:     Compute the Euclidean square distance $d(\vec{a}, \vec{x}_i)$.
 4:     Append $(d(\vec{a}, \vec{x}_i), y_i)$ to *dis*.
 5: **end for**
 6: Select the $k$ smallest entries from *dis* and their corresponding labels to form the $k$ nearest neighbors.
 7: Let $b$ be the majority label among the $k$ nearest neighbors.
 8: **Return** $b$.

---

### 2.2. Multi-Party Computation

MPC enables multiple parties to cooperatively compute a function while keeping their input data private. There are several technical routes of MPC, which include homomorphic encryption-based protocols (Dulek et al., 2016), garbled circuit-based protocols (Ciampi et al., 2021), and secret sharing-based protocols (Ben-Or et al., 2019). In this paper, we employ the two-party additive secret sharing (ASS) (Demmler et al., 2015) and the two-party masked secret sharing (MSS) (Ben-Efraim et al., 2019) as the foundational techniques of Kona.

**Additive Secret Sharing:** To share a secret $x$ to two computation parties $(P_a, P_b)$ by using ASS, two random values $[\![x]\!]_a$ and $[\![x]\!]_b$ are drawn from a ring $\mathbb{Z}_{2^h}$ of size $2^h$, such that $x = [\![x]\!]_a + [\![x]\!]_b (mod\ 2^h)$. Then $[\![x]\!]_a$ is distributed to $P_a$, and $[\![x]\!]_b$ is distributed to $P_b$. To reconstruct $x$ to a user, each $P_i$ $(i \in \{a, b\})$ sends $[\![x]\!]_i$ to the user. Then, the user computes $x = [\![x]\!]_a + [\![x]\!]_b (mod\ 2^h)$.

Throughout this paper, we use the notation $[\![x]\!]$ to indicate that $x$ is shared to the two computation parties by using ASS. Besides, we omit "$(mod\ 2^h)$" in the following sections for simplicity, since all computations are performed on $\mathbb{Z}_{2^h}$.

Let $c$ be a constant value, $[\![x]\!]$ and $[\![y]\!]$ be two additive secret-shared values. In this paper, we leverage the following basic operations, whose implementation is detailed in the literature (Demmler et al., 2015), of ASS.

- **Constant Addition:** $[\![z]\!] = [\![x]\!] + c$, such that $z = x + c$.

- **Constant Multiplication:** $[\![z]\!] = [\![x]\!] * c$, such that $z = x * c$.

- **Share Addition:** $[\![z]\!] = [\![x]\!] + [\![y]\!]$, such that $z = x + y$.

- **Share Multiplication:** $[\![z]\!] = [\![x]\!] * [\![y]\!]$, such that $z = x * y$.

- **Share Comparison:** $[\![z]\!] = ([\![x]\!] < [\![y]\!])$, such that $z = 1$ if $x < y$, otherwise $z = 0$.

- **Share Equality Check:** $[\![z]\!] = ([\![x]\!] == [\![y]\!])$, such that $z = 1$ if $x == y$, otherwise $z = 0$.

**Masked Secret Sharing:** A secret $x$ is said to be shared to two computation parties $P_a$ and $P_b$ by MSS if $P_a$ holds a pair $(U, [\![u]\!]_a)$ and $P_b$ holds a pair $(U, [\![u]\!]_b)$, where $x = U - [\![u]\!]_a - [\![u]\!]_b$.

Throughout this paper, we use the notation $\langle x \rangle$ to indicate that $x$ is shared to the two computation parties $P_a$ and $P_b$ by using MSS, i.e. $\langle x \rangle = (U, [\![u]\!])$.

## 3. Design of Kona

### 3.1. Overview

**Architecture.** As is shown in Figure 1, the architecture of Kona consists of an arbitrary number of data owners (each one denoted as *DO*), an arbitrary number of users (each one denoted as *UR*), and two computation parties (denoted as $P_a$ and $P_b$). Each *DO* (e.g. a hospital) holds a dataset, such as a small set of patient records, and would like to provide KNN classification services with its dataset. Besides, because an individual *DO*'s dataset is often too limited to yield accurate KNN classification results, multiple *DO*s seek to collaborate while preserving the privacy of their dataset. Meanwhile, each *UR* holds its own sensitive data, such as an unlabeled medical record, and seeks a KNN classification result while keeping its data private. $P_a$ and $P_b$ hold no sensitive data but provide computation resources. Note that the computation parties can also be played by *DO* or *UR* if *DO* or *UR* has enough computation resources.

**Workflow.** The workflow of Kona consists of a one-time dataset-share stage, and a KNN-classify stage. In the one-time dataset-share stage, each *DO* secret shares its dataset to $P_a$ and $P_b$ by using MSS. In the KNN-classify stage, each *UR* first secret shares its sensitive data to $P_a$ and $P_b$ using MSS. And then $P_a$ and $P_b$ securely compute the classification result for the sensitive data with our proposed protocols, and send the ASS shares of the classification result to *UR* for reconstruction.

**Security Model.** Kona operates under a semi-honest security model with a dishonest majority. In other words, both

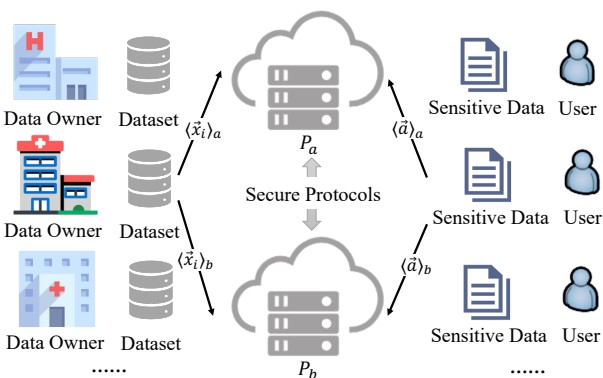

*Figure 1.* Architecture of Kona.

---

**Protocol 1: *DatasetShare***

**Input:** *DO* inputs $n$ samples $\{\vec{x}_i, y_i\}_{i=0}^{n-1}$.
**Output:** $\{\langle \vec{x}_i \rangle = (\vec{U}_i, [\![\vec{u}_i]\!]), [\![y_i]\!]\}_{i=0}^{n-1}$

1: *DO* shares $\{y_i\}_{i=0}^{n-1}$ by using ASS, so that the computation parties hold $\{[\![y_i]\!]\}_{i=0}^{n-1}$.
2: **for** $i = 0$ to $n-1$ in parallel **do**
3:    $P_a$ randomly samples $[\![\vec{u}_i]\!]_a$, and $P_b$ randomly samples $[\![\vec{u}_i]\!]_b$.
4:    $P_a$ sends $[\![\vec{u}_i]\!]_a$ to *DO*, and $P_b$ sends $[\![\vec{u}_i]\!]_b$ to *DO*.
5:    *DO* computes $\vec{U}_i = \vec{x}_i + [\![\vec{u}_i]\!]_a + [\![\vec{u}_i]\!]_b$.
6:    *DO* sends $\vec{U}_i$ to $P_a$ and $P_b$, such that $P_a$ holds $\langle \vec{x}_i \rangle_a = (\vec{U}_i, [\![\vec{u}_i]\!]_a)$ and $P_b$ holds $\langle \vec{x}_i \rangle_b = (\vec{U}_i, [\![\vec{u}_i]\!]_b)$.
7: **end for**

---

computation parties will correctly execute our proposed protocols, yet attempt to infer additional information about the *UR*s' data or the *DO*s' data. Besides, the computation parties are assumed not to collude. Additionally, we assume each *DO* will correctly provide its dataset and each *UR* will correctly provide its sensitive data.

**Data Representation.** We assume the *DO*s hold horizontally distributed datasets, i.e. their datasets contain the same attributes but different samples. Each sample in the datasets is represented by an attribute vector $\vec{x}$ of size $m$ and a label $y$. Besides, the sensitive data held by each *UR* is also represented by an attribute vector $\vec{a}$ of size $m$. We further discuss how to handle vertically distributed datasets in Section 5.

### 3.2. Dataset Share Stage

As is shown in Protocol 1, the *DatasetShare* protocol inputs *DO*'s dataset $\{\vec{x}_i, y_i\}_{i=0}^{n-1}$, and outputs secret-shared datasets to $P_a$ and $P_b$. *DO* first uses ASS to share each label $y_i$ to $P_a$ and $P_b$ (Line 1). Then, for each feature vector $\vec{x}_i$, *DO* uses MSS to share it (Line 2-7). Here, we use MSS to share the attribute vectors rather than ASS, because our proposed Euclidean square distance computation protocol relies on it.

For simplicity and readability, the *DatasetShare* protocol considers only one *DO*. Extending this protocol to support multiple *DO*s is straightforward: simply execute the protocols independently for each additional *DO*. Then, let $P_a$ and $P_b$ merge secret-shared datasets from the *DO*s to get a whole dataset.

### 3.3. KNN Classify Stage

As is shown in Protocol 2, the *KNN-classify* protocol inputs a sample vector $\vec{a}$ from *UR*, and inputs secret-shared attribute vectors, secret-shared labels, and Euclidean triples (introduced in Section 3.4) from $P_a$ and $P_b$. It outputs the predicted label $b$ to *UR*.

In the protocol, *UR* first shares the sample vector $\vec{a}$ by MSS (Line 1-3). Then $P_a$ and $P_b$ leverage the Euclidean triples to efficiently perform Euclidean square distance computation without incurring online communication overhead (Line 4-6). Once all distances are computed, $P_a$ and $P_b$ apply our proposed *DQBubble* protocol (shown in protocol 4) to select the $k$ nearest neighbors, i.e. move the $k$ nearest neighbors to the front of the vectors (Line 7-9). Finally, $P_a$ and $P_b$ obtain the predicted label $[\![b]\!]$ by calling the *LabelCompute* protocol (shown in Appendix B.1), and then send $[\![b]\!]_a$ and $[\![b]\!]_b$ to *UR* for reconstructing $b$ (Line 10-12).

---

**Protocol 2: *KNN-classify***

**Input:** *UR* inputs a sample $\vec{a}$. $P_a$ and $P_b$ inputs $\{\langle \vec{x}_i \rangle = (\vec{U}_i, [\![\vec{u}_i]\!]), [\![y_i]\!]\}_{i=0}^{n-1}$, and Euclidean triples $\{[\![\vec{u}_i]\!]\}_{i=0}^{n-1}, [\![\vec{v}]\!], \{[\![w_i]\!]\}_{i=0}^{n-1}$, where $w_i = (\vec{u}_i - \vec{v})^2$.
**Output:** *UR* obtains the predicted label $b$.

1: $P_a$ sends $[\![\vec{v}]\!]_a$ to *UR*, and $P_b$ sends $[\![\vec{v}]\!]_b$ to *UR*.
2: *UR* computes $\vec{V} = \vec{a} + [\![\vec{v}]\!]_a + [\![\vec{v}]\!]_b$.
3: *UR* sends $\vec{V}$ to $P_a$ and $P_b$, such that $P_a$ holds $\langle \vec{a} \rangle_a = (\vec{V}, [\![\vec{v}]\!]_a)$ and $P_b$ holds $\langle \vec{a} \rangle_b = (\vec{V}, [\![\vec{v}]\!]_b)$.
4: **for** $i = 0$ to $n-1$ **do**
5:    $[\![d_i]\!] = EuclideanDistance(\langle \vec{x} \rangle, \langle \vec{a} \rangle)$.
6: **end for**
7: **for** $j = 0$ to $k-1$ **do**
8:    $\{[\![d_i]\!]\}_{i=j}^{n-1}, \{[\![y_i]\!]\}_{i=j}^{n-1} = DQBubble (\{[\![d_i]\!]\}_{i=j}^{n-1}, \{[\![y_i]\!]\}_{i=j}^{n-1})$.
9: **end for**
10: $[\![b]\!] = LabelCompute(\{[\![y_i]\!]\}_{i=0}^{k-1})$.
11: $P_a$ sends $[\![b]\!]_a$ to *UR*, and $P_b$ sends $[\![b]\!]_b$ to *UR*,.
12: *UR* computes $b = [\![b]\!]_a + [\![b]\!]_b$.

---

### 3.4. Euclidean Square Distance Computation Protocol

**Issue of Existing Methods.** Existing methods incur large communication size during the online phase of secure Euclidean square distance computations, because they require communicating encrypted elements or secret shares for each

distance computation. For example, the method proposed by Sun and Yang (Sun & Yang, 2020) employs homomorphic encryption to securely compute Euclidean square distance and requires communicating two encrypted vectors for each distance computation. Though the method proposed by Li et al. (Li et al., 2023) leverages ASS shares of random pairs ($\{r, r^2\}$) generated in the offline phase to reduce the online communication size, this method still requires communicating a secret-shared vector for each distance computation, because the random pairs are not fully compatible with Euclidean square distance computations.

**Main Idea.** To accelerate the online phase of secure Euclidean square distance computation, we introduce a novel Euclidean triple, which is inspired by the input-independent but function-dependent technique (Ben-Efraim et al., 2019) (generate randomness tailored to a specific function before actual inputs to the function are known). Specifically, an Euclidean triple consists of two secret-shared vectors $[\![\vec{u}]\!]$ and $[\![\vec{v}]\!]$, and a secret-shared scalar $[\![w]\!]$, where each element of $\vec{u}$ and $\vec{v}$ is randomly sampled from $\mathbb{Z}_{2^h}$, and $w = (\vec{u} - \vec{v})^2$. Leveraging this triple, we can compute the ASS shares of Euclidean square distance $d = (\vec{x} - \vec{a})^2$ without online communication. Let $\vec{U} = \vec{x} + \vec{u}$ and $\vec{V} = \vec{a} + \vec{v}$. We have:

$$
\begin{aligned}
d &= (\vec{x} - \vec{a})^2 = ((\vec{U} - \vec{u}) - (\vec{V} - \vec{v}))^2 \\
&= (\vec{U} - \vec{V})^2 - 2(\vec{U} - \vec{V})(\vec{u} - \vec{v}) + (\vec{u} - \vec{v})^2 \\
&= (\vec{U} - \vec{V})^2 - 2(\vec{U} - \vec{V})([\![\vec{u}]\!]_a - [\![\vec{v}]\!]_a) \\
&\quad - 2(\vec{U} - \vec{V})([\![\vec{u}]\!]_b - [\![\vec{v}]\!]_b) + [\![w]\!]_a + [\![w]\!]_b
\end{aligned}
$$

Since $\vec{U}$ has been obtained in the dataset-share stage, $[\![\vec{v}]\!]$ has been obtained from $UR$ (Line 3 in Protocol 2), and the Euclidean triples can be generated in the offline phase, $P_a$ and $P_b$ can locally compute their ASS shares of $d$ by $[\![d]\!]_a = (\vec{U} - \vec{V})^2 - 2(\vec{U} - \vec{V})([\![\vec{u}]\!]_a - [\![\vec{v}]\!]_a) + [\![w]\!]_a$, and $[\![d]\!]_b = -2(\vec{U} - \vec{V})([\![\vec{u}]\!]_b - [\![\vec{v}]\!]_b) + [\![w]\!]_b$.

As is shown in Protocol 3, the *EuclideanDistance* protocol inputs the masked secret-shared two vectors $\langle\vec{x}\rangle = (\vec{U}, [\![\vec{u}]\!])$ and $\langle\vec{a}\rangle = (\vec{V}, [\![\vec{v}]\!])$, as well as a additive secret-shared value $[\![w]\!]$, where $w = (\vec{u} - \vec{v})^2$, and outputs an additive secret-shared value $[\![d]\!]$, where $d = (\vec{x} - \vec{a})^2$. By leveraging the Euclidean triples, both $P_a$ and $P_b$ independently compute their respective shares of the distance, which significantly reduces communication overhead for secure computation for Euclidean square distance.

**Generation of Euclidean Triples.** The Euclidean triples can be generated in the offline phase either by a trusted third party (e.g. a trusted execution environment) or by using a generation protocol based on homomorphic encryption. Note that the communication overhead for generating Euclidean triples based on homomorphic encryption is the same as the cost of generating ASS shares of random pairs in

---

**Protocol 3: *EuclideanDistance***

**Input:** $\langle\vec{x}\rangle = (\vec{U}, [\![u]\!])$, $\langle\vec{a}\rangle = (\vec{V}, [\![v]\!])$, and $[\![w]\!]$, where $w = (\vec{u} - \vec{v})^2$.
**Output:** $[\![d]\!]$, where $d = (\vec{x} - \vec{a})^2$.

1: $P_a$ computes $[\![d]\!]_a = (\vec{U} - \vec{V})^2 - 2(\vec{U} - \vec{V})([\![\vec{u}]\!]_a - [\![\vec{v}]\!]_a) + [\![w]\!]_a$.
2: $P_b$ computes $[\![d]\!]_b = -2(\vec{U} - \vec{V})([\![\vec{u}]\!]_b - [\![\vec{v}]\!]_b) + [\![w]\!]_b$.

---

the baseline (Li et al., 2023) based on homomorphic encryption. We present the generation protocol in Appendix B.3 and the communication analysis in Appendix F.

### 3.5. Divide-and-Conquer Bubble Protocol

**Issue of Existing Methods.** Existing methods (Li et al., 2023; Liu et al., 2019) require a large number of communication rounds to select the nearest neighbors because they rely on sequential bubble protocols. In these protocols, the nearest neighbor is selected by comparing pairs of elements from the end of the distance list to the front, and swapping elements whenever a smaller value appears behind a larger one. Since these comparisons are performed sequentially, selecting the nearest neighbor in a list of size $l$ demands $O(l)$ communication rounds. This high round complexity severely impacts performance, particularly in wide-area network environments.

---

**Protocol 4: *DQBubble***

**Input:** $\{[\![d_i]\!]\}_{i=0}^{l-1}, \{[\![y_i]\!]\}_{i=0}^{l-1}$.
**Output:** $\{[\![d_i]\!]\}_{i=0}^{l-1}, \{[\![y_i]\!]\}_{i=0}^{l-1}$, where $d_0$ is the smallest element in $\{d_i\}_{i=0}^{l-1}$, and each $y_i$ remains paired with its original $d_i$.

1: **while** $l > 1$ **do**
2:     $mid = \lfloor \frac{l}{2} \rfloor$.
3:     **for** $g = 0$ to $mid - 1$ in parallel **do**
4:         $[\![d_g]\!], [\![y_g]\!], [\![d_{g+mid}]\!], [\![y_{g+mid}]\!] = CompareSwap([\![d_g]\!], [\![y_g]\!], [\![d_{g+mid}]\!], [\![y_{g+mid}]\!])$
5:     **end for**
6:     **if** $l$ is odd **then**
7:         $[\![d_0]\!], [\![y_0]\!], [\![d_{l-1}]\!], [\![y_{l-1}]\!] = CompareSwap([\![d_0]\!], [\![y_0]\!], [\![d_{l-1}]\!], [\![y_{l-1}]\!])$
8:     **end if**
9:     $l = \lfloor \frac{l}{2} \rfloor$
10: **end while**

---

**Main Idea.** To reduce the communication rounds needed to select the nearest neighbor, we present a novel bubble protocol based on a divide-and-conquer strategy. Rather than comparing adjacent elements one by one, our protocol partitions the list into pairs and performs secure comparisons in parallel. Specifically, at each iteration, the list is split into multiple pairs, and each pair of elements is compared and swapped if a smaller value appears behind a larger

one. This process then continues on the candidate minimum elements, until there is only one element. With our proposed bubble protocol, although the communication size remains unchanged, the communication rounds required to select the nearest neighbor are reduced to $O(\log l)$.

As is shown in Protocol 4, the *DQBubble* protocol inputs two additive secret-shared lists of size $l$: $\{[\![d_i]\!]\}_{i=0}^{l-1}$ and $\{[\![y_i]\!]\}_{i=0}^{l-1}$. It synchronously reorders both the distance list and the label list so that $d_0$ becomes the smallest distance in $\{d_i\}_{i=0}^{l-1}$, meanwhile, each label remains paired with its original distance. The protocol proceeds in $O(\log l)$ iterations, where each iteration "bubbles" the smallest distance and its corresponding label to the front half of the lists. At each iteration, a *CompareSwap* protocol (shown in Appendix B.2) is performed for each pair of positions $\left(g,\ g+\lfloor\frac{l}{2}\rfloor\right)$ to swap the smaller distance and its associated label to the lower index (Line 2-5). If $l$ is odd, the last element $[\![d_{l-1}]\!]$ is also compared and potentially swapped with $[\![d_0]\!]$ (Line 6-8). After completing these pairwise comparisons, $l$ is updated to $\lfloor\frac{l}{2}\rfloor$ (Line 9), and the above procedure repeats until $l \leq 1$.

## 4. Evaluation

### 4.1. Experiment Setting

**Implementation:** We implement Kona in *C++*, and use function secret sharing (Boyle et al., 2015) to improve the secure comparison operations for fair comparison with the baseline (Li et al., 2023). Each computation party in Kona is simulated by a separate process with one thread. Besides, We perform the computation of Kona on the ring $\mathbb{Z}_{2^{64}}$.

**Experiment Environment:** We conduct experiments on a Linux server equipped with a 32-core 2.4 GHz Intel Xeon CPU and 128GB of RAM. Note that, since each computation party in Kona is simulated by a separate process with one thread, Kona uses at most two cores during the classification process. As for the network setting, we apply the tc tool[2] to simulate two network settings: one is the WAN setting with a bandwidth of 40 megabit per second (Mbps) and 40ms round-trip time (RTT). The other is the LAN setting with 1024 Mbps and sub-millisecond RTT.

**Datasets:** As is shown in Table 2, we use eight real-world datasets from UCI repository (Kelly et al., 2023).

### 4.2. Accuracy Evaluation

**Baseline:** For accuracy evaluation, we adopt the plaintext KNN algorithm in scikit-learn (Pedregosa et al., 2011), which is a famous open-sourced machine learning library implemented in Python, as the baseline.

---

[2]https://man7.org/linux/man-pages/man8/tc.8.html

*Table 2.* Detailed information of employed datasets. '#Label' means the number of label types. Note that we remove the samples without labels from the original datasets.

| Dataset | #Sample ($n$) | #Attribute ($m$) | #Label |
|---------|---------------|------------------|--------|
| Toxicity | 120 | 1203 | 2 |
| Iris | 150 | 4 | 3 |
| Arcene | 200 | 10000 | 2 |
| PEMS-SF | 440 | 137710 | 3 |
| RNA-seq | 800 | 20532 | 5 |
| Spambase | 4601 | 57 | 2 |
| Mnist | 70000 | 784 | 10 |
| Dota2 Games | 102944 | 115 | 2 |

**Dataset Split:** We split each dataset in Table 2 into a training set and a test set with a ratio of $8:2$.

**Hyperparameter $k$:** We set the hyperparameter $k$ to 5, a commonly used value in KNN-related literature (Li et al., 2023; Zhu et al., 2022; Xu & Klabjan, 2021).

As is shown in Table 3, Kona achieves the same accuracy as scikit-learn across all the datasets. This is because our proposed protocol merely transforms the computation from the plaintext domain to the ciphertext domain without altering the underlying computational semantics.

*Table 3.* Accuracy of Kona vs. scikit-learn on eight real-world datasets.

| Dataset | Kona | scikit-learn |
|---------|------|--------------|
| Toxicity | 0.5714 | 0.5714 |
| Iris | 0.9667 | 0.9667 |
| Arcene | 0.9250 | 0.9250 |
| PEMS-SF | 0.7386 | 0.7386 |
| RNA-seq | 1.0 | 1.0 |
| Spambase | 0.9011 | 0.9011 |
| Mnist | 0.9722 | 0.9722 |
| Dota2 Games | 0.5300 | 0.5300 |

### 4.3. Efficiency Evaluation

**Baseline:** For efficiency evaluation, we adopt SecKNN (Li et al., 2023) as the baseline. SecKNN is the most recent state-of-the-art secure KNN framework, which supports multiple data owners and relies on function secret sharing (Boyle et al., 2015) to improve efficiency. As the code of SecKNN is not publicly available, we implement it in *C++* and perform the computation on the ring $\mathbb{Z}_{2^{64}}$.

**Dataset Split:** We randomly select one sample from each dataset in Table 2 as the query and use all the remaining samples as the training set.

**Hyperparameter $k$:** We first fix the hyperparameter $k = 5$ to demonstrate the efficiency improvement of Kona compared to SecKNN across all the datasets in Table 2. Subsequently, we evaluate the effect of varying $k$ on Kona's efficiency using the Arcene dataset.

*Table 4.* Online runtime (second), communication size (MB), and communication rounds of `Kona` vs. `SecKNN` (Li et al., 2023) with $k = 5$.

| | Framework | Dataset | | | | | | | |
|---|---|---|---|---|---|---|---|---|---|
| | | Toxicity | Iris | Arcene | PEMS-SF | RNA-seq | Spambase | Mnist | Dota2 Games |
| Communication Size | Kona | **0.12** (28.25×) | **0.10** (1.10×) | **0.14** (228.07×) | **0.31** (3121.25×) | **0.57** (462.07×) | **3.31** (2.26×) | **50.39** (18.42×) | **74.11** (3.55×) |
| | SecKNN | 3.39 | 0.11 | 31.93 | 967.59 | 263.38 | 7.50 | 928.46 | 263.53 |
| Communication Round | Kona | **88** (19.10×) | **88** (16.72×) | **88** (22.40×) | **98** (44.61×) | **108** (73.89×) | **138** (333.19×) | **178** (3932.42×) | **178** (5783.21×) |
| | SecKNN | 1681 | 1471 | 1972 | 4372 | 7981 | 45981 | 699971 | 1029412 |
| Runtime in WAN | Kona | **2.11** (16.58×) | **2.09** (14.36×) | **2.22** (20.57×) | **3.02** (79.16×) | **3.24** (66.39×) | **6.94** (135.22×) | **63.05** (229.31×) | **90.54** (232.60×) |
| | SecKNN | 34.99 | 30.02 | 45.68 | 239.09 | 215.12 | 938.46 | 14458.60 | 21060.40 |
| Runtime in LAN | Kona | **0.14** (1.28×) | **0.13** (1.15×) | **0.17** (2.94×) | **0.50** (11.36×) | **0.66** (3.13×) | **3.54** (1.21×) | **54.02** (1.26×) | **79.15** (1.18×) |
| | SecKNN | 0.18 | 0.15 | 0.50 | 5.68 | 2.07 | 4.29 | 68.54 | 94.04 |

As is shown in Table 4, we can conclude as follows:

- In terms of communication overhead, `Kona` significantly reduces both the communication size and communication rounds. Specifically, `Kona` reduces communication size by $1.10\times$ (Iris) up to $3121.25\times$ (PEMS-SF), and reduces communication rounds by $16.7\times$ (Iris) to $5783.21\times$ (Dota2 Games). Notably, `Kona` provides greater benefits in reducing communication rounds as the number of samples increases, and greater benefits in reducing communication size as the number of attributes increases. These empirical results align with the analytical complexity results in Table 1, underscoring the effectiveness of the two communication optimization strategies employed by `Kona`.

- In terms of runtime in the WAN setting, `Kona` achieves $14.36\times$ to $232.60\times$ speedups compared to `SecKNN`, where the performance bottleneck lies in communication, with particularly large gains on large datasets like Mnist ($229.31\times$) and Dota2 Games ($232.60\times$). Especially, `Kona` requires only about 90s to obtain a classification result with the dataset Dota2 Games, which is a moderate-sized dataset in the real world. These results confirm that `Kona` should be well-suited for real-world scenarios, where network bandwidth is usually limited and RTT is usually long.

- Even in the LAN setting, where the performance bottleneck lies in both computation and communication, `Kona` remains up to $11.36\times$ faster on PEMS-SF. These results demonstrate that the optimizations of `Kona` should also be useful for scenarios featuring high bandwidth and low latency.

As is shown in Table 5, we can conclude as follows: Although the performance gains of `Kona` gradually decrease as $k$ increases, the performance gains remain consistently significant even at large $k$ values. For example, at $k = 100$,

`Kona` still achieves a $13.15\times$ reduction in communication size, a $19.18\times$ reduction in communication rounds, and a $16.02\times$ speedup in WAN runtime. These results indicate that the optimizations employed by `Kona` remain robust and effective across varying $k$ value settings.

### 4.4. Ablation Evaluation

In order to further show the effectiveness of the two optimizations in `Kona`, we implement the two optimizations of `Kona` into `SecKNN` (Li et al., 2023). We name `SecKNN` with the optimization based on Euclidean triples as `SecKNN-Triples`, and `SecKNN` with the optimization based on the *DQBubble* protocol as `SecKNN-DQBubble`. We compare the communication size, communication rounds, and online runtime, of these two frameworks and the original `SecKNN` with $k = 5$.

As is shown in Figure 2, we can conclude as follows:

- `SecKNN-Triples` significantly reduces the communication size across all datasets ($1.10\times$ to $3064.94\times$), and significantly reduces runtime in both WAN and LAN settings ($1.00\times$ to $10.16\times$), especially for high-dimension datasets (e.g. PEMS-SF). These results confirm that the optimization with Euclidean triples can significantly enhance efficiency over diverse network settings, and should be much more useful for high-dimensional datasets.

- `SecKNN-DQBubble` significantly reduces communication rounds ($16.16\times$ to $5688.51\times$) without introducing additional communication size, and significantly reduces runtime in the WAN setting ($1.62\times$ to $164.56\times$). These results confirm that the optimization with the *DQBubble* protocol can effectively accelerate secure $k$-NN selection in the WAN settings.

Note that we do not include accuracy comparisons in the ablation evaluation because both of our optimizations have no impact on the classification accuracy. As is shown in

*Table 5.* Online runtime (second), communication size (MB), and communication rounds of `Kona` vs. `SecKNN` (Li et al., 2023) for varying $k$ on the Arcene dataset.

| | Framework | $k$ | | | | | | | | |
|---|---|---|---|---|---|---|---|---|---|---|
| | | 1 | 2 | 5 | 10 | 20 | 40 | 60 | 80 | 100 |
| Communication Size | `Kona` | **0.03** (1062.00×) | **0.06** (531.00×) | **0.14** (228.07×) | **0.28** (114.71×) | **0.56** (57.58×) | **1.12** (29.41×) | **1.64** (20.40×) | **2.14** (15.87×) | **2.62** (13.15×) |
| | `SecKNN` | 31.86 | 31.88 | 31.93 | 32.12 | 32.40 | 32.94 | 33.46 | 33.98 | 34.46 |
| Communication Round | `Kona` | **19** (21.05×) | **37** (21.51×) | **88** (22.40×) | **171** (22.76×) | **333** (22.76×) | **655** (21.92×) | **975** (20.86×) | **1279** (19.95×) | **1559** (19.18×) |
| | `SecKNN` | 400 | 796 | 1972 | 3892 | 7582 | 14362 | 20342 | 25522 | 29902 |
| Runtime in WAN | `Kona` | **0.43** (31.32×) | **0.87** (24.90×) | **2.22** (20.57×) | **4.13** (20.54×) | **8.08** (20.80×) | **15.97** (18.73×) | **23.81** (18.02×) | **31.44** (17.03×) | **38.58** (16.02×) |
| | `SecKNN` | 13.47 | 21.67 | 45.68 | 84.85 | 168.07 | 299.14 | 429.06 | 535.71 | 618.31 |
| Runtime in LAN | `Kona` | **0.04** (8.5×) | **0.07** (5.28×) | **0.17** (2.94×) | **0.33** (2.06×) | **0.65** (1.66×) | **1.34** (1.43×) | **2.12** (1.35×) | **2.94** (1.29×) | **3.78** (1.26×) |
| | `SecKNN` | 0.34 | 0.37 | 0.50 | 0.68 | 1.08 | 1.92 | 2.87 | 3.79 | 4.76 |

Table 3, `Kona` consistently achieves identical accuracy to `scikit-learn` across all datasets when both optimizations are applied. Since the two optimizations are independent of each other, the frameworks with only one optimization (`SecKNN-Triples` and `SecKNN-DQBubble`) should still achieve identical accuracy to `scikit-learn`.

## 5. Discussion

**Security Issue of Existing Top-$k$ Protocol.** We note that the existing shuffle-based top-$k$ protocol (Hou et al., 2023) can be employed to select the $k$ nearest neighbors. However, this protocol may leak private information because it cannot always produce indistinguishable execution views for different distance lists. This security issue is discussed in detail in Appendix D.

**Vertical Distributed Datasets.** `Kona` can be extended to support vertically distributed datasets by performing private set intersection (Chen et al., 2017; Kolesnikov et al., 2017) among the *DO*s. Specifically, if the *DO*s have vertically partitioned datasets, they first use a private set intersection to identify the intersection samples of their datasets. Each *DO* then shares the attributes of these intersection samples with $P_a$ and $P_b$. Finally, $P_a$ and $P_b$ combine secret-shared attributes into a complete dataset. With this complete dataset, $P_a$ and $P_b$ can directly apply the privacy-preserving KNN classification protocols presented in this paper.

**Extension to More Computation Parties.** `Kona` can be extended to support more computation parties. Specifically, the divide-and-conquer bubble protocol relies on a secure compare-and-swap operation, which can be realized under multi-party ASS protocols such as those in (Damgård et al., 2012). Furthermore, our Euclidean square distance computation protocol can also be adapted to multi-party settings, which is shown in Appendix C.

**Adaptation to Other Distance Metrics.** The optimizations in `Kona` can be adapted to save communication overhead

for other distance metrics. Concretely, first, our proposed divide-and-conquer bubble protocol makes no assumption on the distance metrics. Thus, it can be applied to any distance metric. Besides, our proposed Euclidean triple optimization can be applied to other distance metrics with minor adaptations. That is because the core idea of our optimization is to use the input-independent but function-dependent technique (correlated randomness) to reduce online communication, and this idea can apply to any distance metric. For example, for cosine similarity ( $\frac{\vec{x} \cdot \vec{a}}{\|\vec{x}\|\|\vec{a}\|}$ ), whose main communication bottleneck lies in secure dot product computations ($\vec{x} \cdot \vec{a}$, $\vec{x} \cdot \vec{x}$ in $\|\vec{x}\|$, and $\vec{a} \cdot \vec{a}$ in $\|\vec{a}\|$), we can adapt our Euclidean triple to be a dot triple, which comprises two secret-shared vectors and their dot products. By leveraging the dot triple, all secure dot product computations in cosine similarity can be executed without online communication. For Hamming distance, which is usually used in binary data, the Euclidean triple can be directly applied in the binary domain because $(x - a)^2$ equals the mismatch indicator. Thus, the sum of squared differences immediately gives the Hamming distance for binary vectors.

**Future work:** In the future, we will extend `Kona` to support more security models, such as the security model that defends against malicious parties, and more distance metrics.

## 6. Related Work

Over the last decade, many privacy-preserving frameworks (Li et al., 2023; Liu et al., 2019; Sun & Yang, 2020; Rong et al., 2016; Wu et al., 2019; Li et al., 2015; Samanthula et al., 2015; Elmehdwi et al., 2014) for KNN classification or query have been proposed. These frameworks can be broadly divided into two categories, based on whether they reveal private information during the classification or query process. The first category (Sun & Yang, 2020; Rong et al., 2016; Wu et al., 2019) reveals some private information to speed up the KNN protocols. For example, the framework proposed by Sun and Yang (Sun & Yang, 2020) reveals

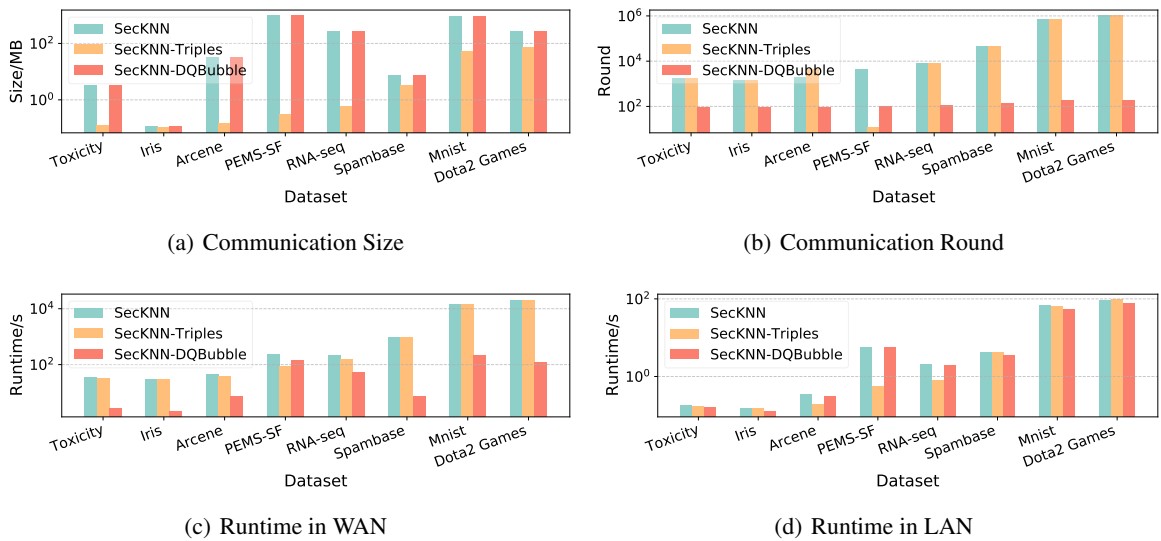

*Figure 2.* Online Runtime (seconds), communication size (MBs), and communication rounds of SecKNN (Li et al., 2023), SecKNN-Triples, and SecKNN-DQBubble with $k = 5$.

which samples are the $k$ nearest neighbors to the data owner, and the framework proposed by Rong et al. (Rong et al., 2016) reveals the indices of the $k$ nearest neighbors. Although the revealed information accelerates the neighbor selection procedure, the information may be exploited to infer the sensitive data. Consequently, frameworks in this category should not be suitable for scenarios where the data owners' and users' data must be strictly protected.

The second category of frameworks (Liu et al., 2019; Cui et al., 2020; Li et al., 2015; 2023; Elmehdwi et al., 2014; Samanthula et al., 2015; Zheng et al., 2024), on the other hand, does not reveal private information during the classification or query process and can be further divided into two subtypes based on how many data owners they can support. (1) Single data owner: Some frameworks (Elmehdwi et al., 2014; Samanthula et al., 2015; Zheng et al., 2024; Cui et al., 2020) can support only one data owner. For example, Elmehdwi et al. (Elmehdwi et al., 2014) employ Paillier homomorphic encryption to realize KNN queries with only one data owner, while Samanthula et al. (Samanthula et al., 2015) extend this approach to support KNN classification. Since these frameworks can only support one data owner, they cannot improve the accuracy of KNN classification by leveraging the data from different data owners. (2) Multiple data owners: The other frameworks (Li et al., 2015; Liu et al., 2019; Li et al., 2023) can support multiple data owners. However, these frameworks usually suffer from computation or communication inefficiency. Specifically, Li et al. (Li et al., 2015) propose an outsourcing framework, which can support multiple data owners. However, this framework suffers from huge computation overhead since it requires $O(n)$ homomorphic operations to compute distance. In contrast,

Liu et al. (Liu et al., 2019) use secret sharing to improve efficiency over homomorphic encryption-based methods, and Li et al.(Li et al., 2023) further reduce communication rounds by integrating function secret sharing. However, these two frameworks require communicating secret shares for each distance computation and incur substantial communication rounds to perform $k$-nearest neighbor selection, which leads to huge communication overhead.

**Our Advantages.** Our proposed framework, Kona, does not reveal private information beyond what can be inferred from the classification result. Besides, Kona supports leveraging data from multiple data owners to improve the accuracy of KNN classification, and meanwhile significantly reduces the communication overhead compared to the state-of-the-art framework.

## 7. Conclusion

In this paper, we propose Kona, an efficient privacy-preservation framework for KNN classification, and optimize the communication overhead from two-fold: (1) We design novel Euclidean triples to eliminate the online communication for secure Euclidean square distance computations. (2) We propose a divide-and-conquer bubble protocol to significantly reduce communication rounds for selecting the $k$ nearest neighbors. Experimental results on eight real-world datasets demonstrate that Kona significantly outperforms the state-of-the-art framework by $1.1\times \sim 3121.2\times$ in communication size, $16.7\times \sim 5783.2\times$ in communication rounds, and $1.1\times \sim 232.6\times$ in runtime.

## Acknowledgments

This paper is supported by National Cryptologic Science Fund of China (2025NCSF01010), Natural Science Foundation of China (92370120, 62172100), and Ant Group Research Fund.

## Impact Statement

This paper presents work whose goal is to advance the field of privacy-preserving machine learning. There are many potential societal consequences of our work, none of which we feel must be specifically highlighted here.

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

## A. Notations

We show the notations used in this paper in Table 6.

Table 6. Notations used in this paper

| Notation | Description |
|----------|-------------|
| $n$ | Number of samples in the dataset |
| $m$ | Number of attributes per sample |
| $k$ | Number of nearest neighbors |
| $\vec{x}_i$ | Attribute vector of the $i$-th sample |
| $y_i$ | Class label of the $i$-th sample |
| $\vec{a}$ | A sample vector |
| $[\![x]\!]$ | Additive secret share of $x$ |
| $\langle x \rangle$ | Masked secret share of $x$ |
| $DO$ | A data owner |
| $UR$ | A query user |
| $P_a$ and $P_b$ | The computation parties |

## B. Remain Protocols for KNN Classification

### B.1. Label Computation

We follow the method proposed by Li et al, (Li et al., 2023) to compute the most common label among the $k$-nearest neighbors.

As is shown in Protocol 5, the *LabelCompute* protocol inputs a list of additive secret-shared labels $\{[\![y_i]\!]\}_{i=0}^{k-1}$, and outputs a single additive secret-shared label $[\![b]\!]$, where $b$ is the most common label among $\{y_i\}_{i=0}^{k-1}$. In this protocol, $P_a$ and $P_b$ first compute the occurrence times for each label (Line 1–6), which results in a counter $[\![cnt_i]\!]$ for each $[\![y_i]\!]$. $P_a$ and $P_b$ then apply the $DQBubble'$ protocol to reorder both the counters $\{[\![cnt_i]\!]\}$ and their associated labels $\{[\![y_i]\!]\}$ so that the label with the highest count is moved to index 0 (Line 7). Note that $DQBubble'$ is a variant of $DQBubble$ to move the maximum value (instead of the minimum value) to the front. Finally, $P_a$ and $P_b$ set $[\![b]\!] = [\![y_0]\!]$ (Line 8).

---

**Protocol 5: *LabelCompute***

**Input:** $\{[\![y_i]\!]\}_{i=0}^{k-1}$.
**Output:** $[\![b]\!]$, where $b$ is the most common label in $\{y_i\}_{i=0}^{k-1}$.

1: **for** $i = 0$ to $k - 1$ in parallel **do**
2:    $[\![cnt_i]\!] = [\![0]\!]$.
3:    **for** $j = 0$ to $k - 1$ in parallel **do**
4:       $[\![cnt_i]\!] = [\![cnt_i]\!] + ([\![y_j]\!] == [\![y_i]\!])$.
5:    **end for**
6: **end for**
7: $\{[\![cnt_i]\!]\}_{i=0}^{k-1}, \{[\![y_i]\!]\}_{i=0}^{k-1} = DQBubble'$
   $(\{[\![cnt_i]\!]\}_{i=0}^{k-1}, \{[\![y_i]\!]\}_{i=0}^{k-1})$.
8: $[\![b]\!] = [\![y_0]\!]$.

---

### B.2. Compare and Swap

As is shown in Protocol 6, the *CompareSwap* protocol inputs four additive secret-shared values $[\![a_0]\!], [\![b_0]\!], [\![a_1]\!], [\![b_1]\!]$, and outputs four updated additive secret-shared values $[\![a_0']\!], [\![b_0']\!], [\![a_1']\!], [\![b_1']\!]$. If $[\![a_0]\!] < [\![a_1]\!]$, the outputs remain in the same order; otherwise, the protocol swaps them. First, $P_a$ and $P_b$ perform a secure comparison $[\![comp]\!] = ([\![a_0]\!] < [\![a_1]\!])$ to determine which of the two values is smaller (Line 1). Next, $P_a$ and $P_b$ update the shares of $a$ and $b$ based on $[\![comp]\!]$. If $[\![comp]\!] = 1$, indicating $[\![a_0]\!] < [\![a_1]\!]$, then $[\![a_0']\!] = [\![a_0]\!]$ and $[\![a_1']\!] = [\![a_1]\!]$; otherwise, the two are swapped (Line 2-3). The same logic applies to $[\![b_0]\!]$ and $[\![b_1]\!]$ (Line 4-5).

---

**Protocol 6: *CompareSwap***

**Input:** $[\![a_0]\!], [\![b_0]\!], [\![a_1]\!], [\![b_1]\!]$.
**Output:** $[\![a_0']\!], [\![b_0']\!], [\![a_1']\!], [\![b_1']\!]$, such that $a_0' = a_0$, $b_0' = b_0$, $a_1' = a_1$ and $b_1' = b_1$ if $a_0 < a_1$, else $a_0' = a_1$, $b_0' = b_1$, $a_1' = a_0$ and $b_1' = b_0$.

1: $[\![comp]\!] = [\![a_0]\!] < [\![a_1]\!]$.
2: $[\![a_0']\!] = [\![a_0]\!] * [\![comp]\!] + (1 - [\![comp]\!]) * [\![a_1]\!]$.
3: $[\![a_1']\!] = [\![a_0]\!] + [\![a_1]\!] - [\![a_0']\!]$.
4: $[\![b_0']\!] = [\![b_0]\!] * [\![comp]\!] + (1 - [\![comp]\!]) * [\![b_1]\!]$.
5: $[\![b_1']\!] = [\![b_0]\!] + [\![b_1]\!] - [\![b_0']\!]$.

---

### B.3. Euclidean Triples Generation

Note that $P_a$ and $P_b$ have obtained $\{[\![u_i]\!]\}_{i=0}^{n-1}$ in the dataset-share stage. As long as $DO$'s dataset remains unchanged, these values do not need to be regenerated. Consequently, producing new Euclidean triples only requires generating an additive secret-shared vector $[\![\vec{v}]\!]$ and computing the corresponding $\{[\![w_i]\!]\}_{i=0}^{n-1}$. This approach reduces computation and communication overhead for generating Euclidean triples, especially for scenarios where the dataset is stable.

As is shown in Protocol 7, the *EuclideanTriples* protocol inputs additive secret-shared vectors $\{[\![\vec{u}_i]\!]\}_{i=0}^{n-1}$, and outputs additive secret-shared Euclidean triples $\{[\![\vec{u}_i]\!]\}_{i=0}^{n-1}$, $[\![\vec{v}]\!]$, and $\{[\![w_i]\!]\}_{i=0}^{n-1}$, where $w_i = (\vec{u}_i - \vec{v})^2$. To begin, $P_a$ and $P_b$ each sample their share of $\vec{v}$ (Line 1). Since $w_i = (\vec{u}_i - \vec{v})^2 = [([\![u_i]\!]_a - [\![v]\!]_a) + ([\![u_i]\!]_b - [\![v]\!]_b)]^2$, the computation expands into three terms: $([\![u_i]\!]_a - [\![v]\!]_a)^2$, $([\![u_i]\!]_b - [\![v]\!]_b)^2$, and the cross term $2([\![u_i]\!]_a - [\![v]\!]_a)([\![u_i]\!]_b - [\![v]\!]_b)$. Here, $([\![u_i]\!]_a - [\![v]\!]_a)^2$ can be locally computed by $P_a$, and $([\![u_i]\!]_b - [\![v]\!]_b)^2$ can be locally computed by $P_b$. The primary challenge is to securely compute the cross term. To achieve this, $P_a$ encrypts $([\![u_i]\!]_a - [\![v]\!]_a)$ and sends the ciphertext to $P_b$ (Line 3-4). Then $P_b$ uses homomorphic operations to compute $2([\![u_i]\!]_a - [\![v]\!]_a)([\![u_i]\!]_b - [\![v]\!]_b)$ under encryption, adds a random mask $r_i$, and returns the masked result to $P_a$ (Line 5-6). After decrypting, $P_a$ obtains the masked cross term and computes $[\![w_i]\!]_a = ([\![u_i]\!]_a - [\![v]\!]_a)^2 + 2([\![u_i]\!]_a - [\![v]\!]_a)([\![u_i]\!]_b - [\![v]\!]_b) + r_i$, while

$P_b$ computes $[\![w_i]\!]_b = ([\![u_i]\!]_b - [\![v]\!]_b)^2 - r_i$ (Line 7-8).

Note that the communication overhead of this protocol is the same as the cost of generating ASS shares of random pairs in the baseline (Li et al., 2023) based on homomorphic encryption (see Appendix F),.

---

**Protocol 7: *EuclideanTriples***

**Input:** $\{[\![\vec{u}_i]\!]\}_{i=0}^{n-1}$.
**Output:** Euclidean triples $\{[\![\vec{u}_i]\!]\}_{i=0}^{n-1}$, $[\![\vec{v}]\!]$, $\{[\![w_i]\!]\}_{i=0}^{n-1}$, where $w_i = (\vec{u}_i - \vec{v})^2$.

1: $P_a$ randomly samples $[\![\vec{v}]\!]_a$, and $P_b$ randomly samples $[\![\vec{v}]\!]_b$.
2: **for** $i = 0$ to $n - 1$ in parallel **do**
3:    $P_a$ locally computes $tmp_i = [\![\vec{u}_i]\!]_a - [\![\vec{v}]\!]_a$.
4:    $P_a$ uses its private key to encrypt $tmp_i$ to get $Enc(tmp_i)$, and sends $Enc(tmp_i)$ to $P_b$.
5:    $P_b$ randomly samples $r_i$ and computes $Enc(tmp2_i) = 2 * Enc(tmp_i)([\![\vec{u}_i]\!]_b - [\![\vec{v}]\!]_b) + r_i$.
6:    $P_b$ sends $Enc(tmp2_i)$ to $P_a$.
7:    $P_a$ decrypts $Enc(tmp2_i)$ to get $tmp2_i$.
8:    $P_a$ computes $[\![w_i]\!]_a = ([\![\vec{u}_i]\!]_a - [\![\vec{v}]\!]_a)^2 + tmp2_i$, and $P_b$ computes $[\![w_i]\!]_b = ([\![\vec{u}_i]\!]_b - [\![\vec{v}]\!]_b)^2 - r_i$.
9: **end for**

---

## C. Euclidean Square Distance Computation for More Computation Parties

In this section, we present the protocols to compute Euclidean square distance in multiple computation parties. Here, we assume the number of computation parties is $\mathcal{N}$. The computation parties are defined as $P_0$ to $P_{\mathcal{N}-1}$. $[\![x]\!]$ represents $x$ is additive secret shared among $\mathcal{N}$ computation parties, i.e. each $P_i$ holds $[\![x]\!]_i$, where $x = \sum[\![x]\!]_i$.

As is shown in Protocol 8, the *NPartyEuclideanDistance* protocol securely computes the Euclidean square distance in an $\mathcal{N}$-party setting. This protocol inputs masked secret-shared vectors $\langle\vec{x}\rangle = (\vec{U}, [\![\vec{u}]\!])$, $\langle\vec{a}\rangle = (\vec{V}, [\![\vec{v}]\!])$, and the additive secret-shared scalar $[\![w]\!]$, where $w = (\vec{u} - \vec{v})^2$, and outputs the additive secret-shared square distance $[\![d]\!]$ ($d = (x - a)^2$). In this protocol, party $P_0$ locally computes $[\![d]\!]_0 = (\vec{U} - \vec{V})^2 - 2(\vec{U} - \vec{V})([\![\vec{u}]\!]_0 - [\![\vec{v}]\!]_0) + [\![w]\!]_0$ (Line 1), while each other party $P_j$ for $j \in [1, \mathcal{N} - 1]$ computes $[\![d]\!]_j = -2(\vec{U} - \vec{V})([\![\vec{u}]\!]_j - [\![\vec{v}]\!]_j) + [\![w]\!]_j$ (Line 2). Note that $\sum_{j=0}^{\mathcal{N}-1}[\![d]\!]_j = (\vec{U} - \vec{V})^2 - 2 * (\vec{U} - \vec{V})(\vec{u} - \vec{v}) + (\vec{u} - \vec{v})^2 = ((\vec{U} - \vec{u}) - (\vec{V} - \vec{v})^2 = (\vec{x} - \vec{a})^2$, thus completing the secure Euclidean square distance computation in a multi-party setting.

As is shown in Protocol 9, the *NPartyEuclideanTriples* protocol generates Euclidean triples in an $\mathcal{N}$-party setting. This protocol inputs additive secret-shared vectors $\{[\![\vec{u}_i]\!]\}_{i=0}^{n-1}$, and outputs additive secret-shared Eu-

---

**Protocol 8: *NPartyEuclideanDistance***

**Input:** $\langle\vec{x}\rangle = (\vec{U}, [\![u]\!])$, $\langle\vec{a}\rangle = (\vec{V}, [\![v]\!])$, and $[\![w]\!]$, where $w = (\vec{u} - \vec{v})^2$.
**Output:** $[\![d]\!]$, where $d = (\vec{x} - \vec{a})^2$.

1: $P_0$ computes $[\![d]\!]_0 = (\vec{U} - \vec{V})^2 - 2(\vec{U} - \vec{V})([\![\vec{u}]\!]_0 - [\![\vec{v}]\!]_0) + [\![w]\!]_0$.
2: $P_j$ computes $[\![d]\!]_j = -2(\vec{U} - \vec{V})([\![\vec{u}]\!]_j - [\![\vec{v}]\!]_j) + [\![w]\!]_j$ for $j \in [1, \mathcal{N} - 1]$.

---

clidean triples ($\{[\![\vec{u}_i]\!]\}_{i=0}^{n-1}$, $[\![\vec{v}]\!]$, and $\{[\![w_i]\!]\}_{i=0}^{n-1}$, where $w_i = (\vec{u}_i - \vec{v})^2$). Each party $P_j$ first samples a random vector $[\![\vec{v}]\!]_j$ (Line 1). Then, for each $q \in [0, n - 1]$, party $P_q$ computes a local partial square $[\![w_i]\!]_q = ([\![\vec{u}_i]\!]_q - [\![\vec{v}]\!]_q)^2$ (Line 3-5). The main challenge lies in securely computing the cross term $2([\![\vec{u}_i]\!]_q - [\![\vec{v}]\!]_q)([\![\vec{u}_i]\!]_p - [\![\vec{v}]\!]_p)$ for all pairs of parties $(P_q, P_p)$ ($q \in [0, \mathcal{N} - 1]$, $p \in [q+1, \mathcal{N} - 1]$). To achieve this, $P_q$ locally computes $tmp_{iq} = ([\![\vec{u}_i]\!]_q - [\![\vec{v}]\!]_q)$ and encrypts it using its private key, obtaining $Enc(tmp_{iq})$ (Line 7-8). It then sends $Enc(tmp_{iq})$ to $P_p$, which homomorphically computes $2 * Enc(tmp_{iq})([\![\vec{u}_i]\!]_p - [\![\vec{v}]\!]_p)$ and adds a random mask $r_{iqp}$ (Line 10-12). The result is then sent to $P_q$, and $P_q$ decrypts it to obtain $tmp2_{iqp}$ and incorporates $tmp2_{iqp}$ into $[\![w_i]\!]_q$ (Line 13-14). And $P_p$ updates $[\![w_i]\!]_p$ by subtracting $r_{iqp}$ (Line 15).

---

**Protocol 9: *NPartyEuclideanTriples***

**Input:** $\{[\![\vec{u}_i]\!]\}_{i=0}^{n-1}$.
**Output:** Euclidean triples $\{[\![\vec{u}_i]\!]\}_{i=0}^{n-1}$, $[\![\vec{v}]\!]$, $\{[\![w_i]\!]\}_{i=0}^{n-1}$, where $w_i = (\vec{u}_i - \vec{v})^2$.

1: $P_j$ randomly samples $[\![\vec{v}]\!]_j$ for $j \in [0, \mathcal{N} - 1]$.
2: **for** $i = 0$ to $n - 1$ in parallel **do**
3:    **for** $q = 0$ to $\mathcal{N} - 1$ in parallel **do**
4:      $P_q$ locally computes $[\![w_i]\!]_q = ([\![\vec{u}_i]\!]_q - [\![\vec{v}]\!]_q)^2$.
5:    **end for**
6:    **for** $q = 0$ to $\mathcal{N} - 1$ in parallel **do**
7:      $P_q$ locally computes $tmp_{iq} = [\![\vec{u}_i]\!]_q - [\![\vec{v}]\!]_q$.
8:      $P_q$ uses its private key to encrypt $tmp_{iq}$ to get $Enc(tmp_{iq})$.
9:      **for** $p = q + 1$ to $\mathcal{N} - 1$ in parallel **do**
10:        $P_q$ sends $Enc(tmp_{iq})$ to $P_p$.
11:        $P_p$ randomly samples $r_{iqp}$ and computes $Enc(tmp2_{iqp}) = 2 * Enc(tmp_{iq})([\![\vec{u}_i]\!]_p - [\![\vec{v}]\!]_p) + r_{iqp}$.
12:        $P_p$ sends $Enc(tmp2_{iqp})$ to $P_q$.
13:        $P_q$ decrypts $Enc(tmp2_{iqp})$ to get $tmp2_{iqp}$.
14:        $P_q$ computes $[\![w_i]\!]_q = [\![w_i]\!]_q + tmp2_{iqp}$.
15:        $P_p$ computes $[\![w_i]\!]_p = [\![w_i]\!]_p - r_{iqp}$.
16:      **end for**
17:    **end for**
18: **end for**

---

## D. Security Issue in Existing Top-K Protocol

In this section, we discuss the security issue in the existing shuffle-based top-k protocol (Hou et al., 2023).

As is shown in Protocol 10, the *Top-K* protocol inputs an additive secret-shared list of $n$ distances $\{[\![d_i]\!]\}_{i=0}^{n-1}$, and outputs $k$ additive secret-shared distances $\{[\![d_i']\!]\}_{i=0}^{k-1}$ that correspond to the $k$ minimum values. To begin with, $P_a$ and $P_b$ shuffles the input list $\{[\![d_i]\!]\}_{i=0}^{n-1}$ using a secure shuffle protocol (Chase et al., 2020) (Line 1). Then, $P_a$ and $P_b$ recursively partition the shuffled list into a left subset $S_L$ and a right subset $S_R$ using a pivot element $[\![\text{pivot}]\!] = [\![x_0]\!]$ (Line 2-12). For each $[\![x_i]\!]$, a secure comparison $[\![x_i]\!] < [\![\text{pivot}]\!]$ is performed (Line 7). Then, $P_a$ and $P_b$ reveal $[\![b_i]\!]$ (the comparison bit) to place $[\![x_i]\!]$ into either $S_L$ or $S_R$ (Line 9-13). The protocol proceeds recursively: if $K' = |S_L|$ equals $K$, $S_L$ itself is returned (Line 16-17); if $K' < K$, then elements from $S_R$ must also be included (Line 18-19); if $K' > K$, the algorithm recurses on $S_L$ to select only $K$ elements (Line 20-21).

The *Top-K* protocol is insecure because it cannot always produce the same views for different distance lists. Consider two distance lists: where all distances are identical versus where the distances vary. An adversary (such as $P_a$) observing the protocol execution can distinguish between these two lists. Specifically, if all distances are identical, when comparing each $d_i$ with the pivot, the comparison always returns 'false'. Consequently, the subset $S_R$ is always empty. If the distances are not equal, the comparisons would yield a mix of true and false results, thus splitting the elements into distinct subsets $S_L$ and $S_R$ at every pivot. The resulting partition pattern would differ significantly from the case where all distances are identical. Thus, by observing the partition pattern, the adversary can distinguish between these two lists.

## E. Security Analysis

We analyze the security of our protocols using the universal composability (UC) theorem (UC-secure protocols can be composed arbitrarily without compromising overall security) and the standard real/ideal world paradigm. We consider the two computation parties $P_a$ and $P_b$, the data owner (*DO*), and the user (*UR*) involved in Kona are semi-honest. Besides, we assume that $P_a$ and $P_b$ will not collude with each other. In other words, we assume there exists an adversary who can corrupt *DO*, *UR*, and one of the two computation parties, and follows the protocols but attempts to learn additional information from the protocol transcripts. Our goal is to show that our proposed protocols reveal no information beyond what can be deduced from the intended outputs to the adversary.

For each protocol, we first define an ideal functionality

---

**Protocol 10: *Top-K***

**Input:** $\{[\![d_i]\!]\}_{i=0}^{n-1}$.
**Output:** $\{[\![d_i']\!]\}_{i=0}^{k-1}$ with $\{d_i'\}_{i=0}^{k-1}$ being the $K$ minimum values of $\{d_i\}_{i=0}^{n-1}$.

1: $\{[\![d_i']\!]\}_{i=0}^{n-1} = \text{Shuffle}(\{[\![d_i]\!]\}_{i=0}^{n-1})$.
2: $\{[\![d_i']\!]\}_{i=0}^{k-1} = \text{select}(\{[\![d_i']\!]\}_{i=0}^{n-1}, K)$.
3: **Function** $\text{select}(\{[\![x_i]\!]\}_{i=0}^{l-1}, K)$:
4:     $[\![\text{pivot}]\!] := [\![x_0]\!]$
5:     $S_L := \{\}, S_R := \{[\![\text{pivot}]\!]\}$
6:     **For** $i := 1$ to $l - 1$
7:         $\{[\![b_i]\!]\} = [\![x_i]\!] < [\![\text{pivot}]\!]$
8:         $P_a$ and $P_b$ reveal $[\![b_i]\!]$ and get $b_i$.
9:     **if** $b_i = 0$
10:         $S_L := S_L \cup \{[\![x_i]\!]\}$
11:     **else**
12:         $S_R := S_R \cup \{[\![x_i]\!]\}$
13:     **end if**
14:     **End For**
15:     $K' = sizeof(S_L)$.
16:     **if** $K' == K$
17:         **return** $S_L$
18:     **if** $K' < K$
19:         **return** $S_L \cup select(S_R, K - K')$
20:     **if** $K' > K$
21:         **return** $select(S_L, K)$
22: **End Function**

---

capturing the inputs and outputs as follows.

- $\mathcal{F}_{\text{DatasetShare}}$: $\mathcal{F}_{\text{DatasetShare}}$ receives the dataset $\{(\vec{x}_i, y_i)\}$ from *DO* and provides MSS shares of attributes and ASS shares of labels to $P_a$ and $P_b$.

- $\mathcal{F}_{\text{KNN-classify}}$: $\mathcal{F}_{\text{KNN-classify}}$ receives MSS shares of attributes, ASS shares of labels, Euclidean triples from $P_a$ and $P_b$, and a sample $\vec{a}$ from *UR*, and provides the classification label $b$ to *UR*.

- $\mathcal{F}_{\text{EuclideanDistance}}$: $\mathcal{F}_{\text{EuclideanDistance}}$ receives MSS shares of two vectors from $P_a$ and $P_b$, and provides ASS shares of their Euclidean square distance to $P_a$ and $P_b$.

- $\mathcal{F}_{\text{DQBubble}}$: $\mathcal{F}_{\text{DQBubble}}$ receives ASS shares of a distance list and a label list from $P_a$ and $P_b$ and provides the ASS shares of an updated distance list and an updated label list to $P_a$ and $P_b$, where the smallest distance at the front and each label remains paired with its original distance.

- $\mathcal{F}_{\text{LabelCompute}}$: $\mathcal{F}_{\text{LabelCompute}}$ receives ASS shares of labels from $P_a$ and $P_b$, and provides the ASS shares of the label that occurs most frequently to $P_a$ and $P_b$.

- $\mathcal{F}_{\text{CompareSwap}}$: $\mathcal{F}_{\text{CompareSwap}}$ receives ASS shares of two pairs $(a_0, b_0)$ and $(a_1, b_1)$ from $P_a$ and $P_b$, and provides ASS shares of the new pairs $(a_0', b_0')$ and $(a_1', b_1')$ to $P_a$ and $P_b$, where the output pairs are ordered

such that if $a_0 < a_1$, then $(a'_0, b'_0) = (a_0, b_0)$ and $(a'_1, b'_1) = (a_1, b_1)$; otherwise, $(a'_0, b'_0) = (a_1, b_1)$ and $(a'_1, b'_1) = (a_0, b_0)$.

- $\mathcal{F}_{\text{EuclideanTriples}}$: $\mathcal{F}_{\text{EuclideanTriples}}$ receives ASS shares of $\{\vec{u}_i\}_{i=0}^{n-1}$ from $P_a$ and $P_b$, and provides ASS shares of $\vec{v}$ and $\{w_i\}_{i=0}^{n-1}$ to $P_a$ and $P_b$, where $w_i = (\vec{u}_i - \vec{v})^2$.

We then prove the security of the protocols by using the UC theorem or constructing a simulator $\mathcal{S}$ that interacts with the ideal functionalities and produces a simulated view indistinguishable from the adversary's view in the real protocol execution. In all descriptions below, unless otherwise specified, $i$ ranges from 0 to $n - 1$.

***DatasetShare* (Protocol 1).** In this protocol, *DO* first sends $[\![y_i]\!]_a$ to $P_a$, and sends $[\![y_i]\!]_b$ to $P_b$. *DO* receives random shares $[\![\vec{u}_i]\!]_a$ and $[\![\vec{u}_i]\!]_b$ from $P_a$ and $P_b$, and directly sends $U_i = x_i + [\![\vec{u}_i]\!]_a + [\![\vec{u}_i]\!]_b$ to $P_a$ and $P_b$. Because the adversary can corrupt *DO* and one of $P_a$ and $P_b$ in this protocol. We provide the simulations for the following three cases:

Case 1: The adversary corrupts *DO*. $\mathcal{S}$ proceeds as follows:

1. $\mathcal{S}$ receives the raw dataset $\{\vec{x}_i, y_i\}$ from $\mathcal{F}_{\text{DatasetShare}}$.

2. $\mathcal{S}$ locally generates random values $[\![\vec{u}_i]\!]_a$ (as if receives from $P_a$) and $[\![\vec{u}_i]\!]_b$ (as if receives from $P_b$).

Case 2: The adversary corrupts $P_a$. $\mathcal{S}$ proceeds as follows:

1. $\mathcal{S}$ locally generates random values $[\![y_i]\!]_a$ and $U_i$ (as if receives from *DO*).

Case 3: The adversary corrupts *DO* and $P_a$: $\mathcal{S}$ proceeds as follows:

1. $\mathcal{S}$ receives the raw dataset $\{\vec{x}_i, y_i\}$ from $\mathcal{F}_{\text{DatasetShare}}$.

2. $\mathcal{S}$ locally generates random value $[\![\vec{u}_i]\!]_b$ (as if receives from $P_b$).

Because the protocols are symmetric, the simulations for the cases where the adversary corrupts $P_b$ follow similarly.

In all the above cases, from the adversary's perspective, the values it receives in the real protocol are identical in distribution to the values it receives in the ideal world. Consequently, the *DatasetShare* protocol is secure against the adversary who can corrupt *DO*, *UR*, and one of $P_a$ and $P_b$.

***KNN-classify* (Protocol 2).** In this protocol, the steps shown in Line 4 to Line 10 totally rely on sub-protocols, which are proven secure below. Hence, these steps are secure under the UC theorem. Except for these steps, $P_a$ sends $[\![v]\!]_a$ to *UR*, and $P_b$ sends $[\![v]\!]_b$ to *UR* (Line 1). Then, *UR* computes

$\vec{V} = \vec{a} + [\![v]\!]_a + [\![v]\!]_b$, and sends $\vec{V}$ to $P_a$ and $P_b$ (Line 2-3). Finally, $P_a$ sends $[\![b]\!]_a$ to *UR*, and $P_b$ sends $[\![b]\!]_b$ to *UR* (Line 11). Because the adversary can corrupt *UR* and one of $P_a$ and $P_b$ in this protocol. We provide the simulations for the following three cases:

Case 1: The adversary corrupts *UR*. $\mathcal{S}$ proceeds as follows:

1. $\mathcal{S}$ receives the raw sample $\vec{a}$ from $\mathcal{F}_{\text{KNN-Classify}}$.

2. $\mathcal{S}$ locally generates random values $[\![v]\!]_a$ (as if receives from $P_a$) and $[\![v]\!]_b$ (as if receives from $P_b$).

3. $\mathcal{S}$ receives the final predicted label $b$ from $\mathcal{F}_{\text{KNN-Classify}}$, and local generated random values $[\![b]\!]_a$ (as if receives from $P_a$) and $[\![b]\!]_b$ (as if receives from $P_b$), where $[\![b]\!]_a + [\![b]\!]_b = b$.

Case 2: The adversary corrupts $P_a$. $\mathcal{S}$ proceeds as follows:

1. $\mathcal{S}$ locally generates random value $\vec{V}$ (as if receives from *UR*).

Case 3: The adversary corrupts *UR* and $P_a$: $\mathcal{S}$ proceeds as follows:

1. $\mathcal{S}$ receives the raw sample $\vec{a}$ from $\mathcal{F}_{\text{KNN-Classify}}$.

2. $\mathcal{S}$ locally generates random values $[\![v]\!]_b$ (as if receives from $P_b$).

3. $\mathcal{S}$ receives the final predicted label $b$ from $\mathcal{F}_{\text{KNN-Classify}}$, and local generated random values $[\![b]\!]_a$ and $[\![b]\!]_b$ (as if receives from $P_b$), where $[\![b]\!]_a + [\![b]\!]_b = b$.

In all the above cases, from the adversary's perspective, the values it receives in the real protocol are identical in distribution to the values it receives in the ideal world. Consequently, the *KNN-classify* protocol is secure against the adversary who can corrupt *DO*, *UR*, and one of $P_a$ and $P_b$.

***EuclideanDistance* (Protocol 3).** In this protocol, $P_a$ locally computes $[\![d]\!]_a = (\vec{U} - \vec{V})^2 - 2(\vec{U} - \vec{V})([\![\vec{u}]\!]_a - [\![\vec{v}]\!]_a) + [\![w]\!]_a$, and $P_b$ computes $[\![d]\!]_b = -2(\vec{U} - \vec{V})([\![\vec{u}]\!]_b - [\![\vec{v}]\!]_b) + [\![w]\!]_b$. In the case of $P_a$ being corrupted. $\mathcal{S}$ proceeds as follows:

1. $\mathcal{S}$ receives $\vec{U}, [\![u]\!]_a, \vec{V}, [\![v]\!]_a$, and $[\![w]\!]$, where $w = (\vec{u} - \vec{v})^2$, from $\mathcal{F}_{\text{EuclideanDistance}}$.

2. $\mathcal{S}$ locally computes $[\![d]\!]_a = (\vec{U} - \vec{V})^2 - 2(\vec{U} - \vec{V})([\![\vec{u}]\!]_a - [\![\vec{v}]\!]_a) + [\![w]\!]_a$.

***DQBubble* (Protocol 4).** This protocol totally relies on the composition of *CompareSwap* protocol, which is proven

secure below. Hence, this protocol is secure under the UC theorem.

*LabelCompute* (**Protocol 5**). This protocol totally relies on the composition of the secure equality test and *DQBubble'* protocol, which is proven secure. Hence, this protocol is secure under the UC theorem.

*CompareSwap* (**Protocol 6**). This protocol totally relies on the composition of secure comparison and secure multiplication. Hence, this protocol is secure under the UC theorem.

*EuclideanTriples* (**Protocol 7**). In this protocol, $P_a$ sends $Enc(tmp_i)$ to $P_b$ (Line 4), and $P_b$ sends $Enc(tmp2_i)$ to $P_a$. In the case of $P_a$ being corrupted. $\mathcal{S}$ randomly generate $Enc(tmp2_i)$ (as if received from $P_b$). In the case of $P_b$ being corrupted. $\mathcal{S}$ randomly generate $Enc(tmp_i)$ (as if received from $P_a$). In all the above cases, from the adversary's perspective, the values it receives in the real protocol are identical in distribution to the values it receives in the ideal world. Consequently, the *EuclideanTriples* protocol is secure against the adversary who can corrupt *DO*, *UR*, and one of $P_a$ and $P_b$.

## F. Communication Analysis

In this section, we first analyze the communication overhead of each protocol in Kona, then compare the communication overhead for generating Euclidean triples and generating ASS shares of random pairs (Li et al., 2023). We assume that each of the basic operations described in Section 2.2 has an $O(1)$ communication size and can be completed in $O(1)$ communication rounds.

- *DatasetShare* (**Protocol 1**): In this protocol, *DO* shares $n$ samples of dimension $m$ by MSS, and shares $n$ label by ASS. Hence, the total communication overhead is $O(n * m)$ communication size in $O(1)$ communication rounds.

- *KNN-classify* (**Protocol 2**): In this protocol, *UR* shares a new sample $\vec{a}$ by MSS and then receives the final label. Besides, the *EuclideanDistance* protocol involves no online communication, executing the *DQBubble* protocol k times requires exchanging $O(n)$ secret-shared values in $O(k \log n)$ communication rounds, and the *LabelCompute* protocol requires $O(k^2)$ communication size in $O(\log k)$ communication rounds. . Therefore, the total communication overhead is $O(kn)$ communication size in $O(k \log n)$ communication rounds.

- *EuclideanDistance* (**Protocol 3**): In this protocol, $P_a$ and $P_b$ compute locally. Hence, the communication overhead of this protocol is zero.

- *DQBubble* (**Protocol 4**): In this protocol, $P_a$ and $P_b$ select

the smallest element in a list of size $l$ in $O(\log l)$ iterations. In each iteration, the size of the list is halved, and $P_a$ and $P_b$ compare and swap each pair in parallel. Therefore, the communication overhead is $O(l)$ communication size in $O(\log l)$ communication rounds.

- *LabelCompute* (**Protocol 5**): In this protocol, $P_a$ and $P_b$ compute the occurrence count for each label through parallel comparisons and additions. Specifically, for each of the $k$ labels, $P_a$ and $P_b$ compare against all others, resulting in $k^2$ operations that can be performed in parallel. This phase incurs an overall communication overhead of $O(k^2)$ communication size in $O(1)$ communication rounds. Subsequently, $P_a$ and $P_b$ invoke the *DQBubble'* protocol on the list of $k$ counters and associated labels to reorder them such that the label with the highest count is positioned at the front. The *DQBubble'* protocol requires $O(k)$ communication size in $O(\log k)$ rounds. Therefore, by combining both phases, the total communication overhead is $O(k^2)$ in $O(\log k)$ communication rounds.

- *CompareSwap* (**Protocol 6**): In this protocol, $P_a$ and $P_b$ securely compare two additive secret-shared values $[\![a_0]\!]$ and $[\![a_1]\!]$ and swap them if $[\![a_0]\!] > [\![a_1]\!]$. Hence, the communication overhead is $O(1)$ communication size in $O(1)$ communication rounds.

- *EuclideanTriples* (**Protocol 7**): This protocol is executed in the offline phase to generate Euclidean triples. In this protocol, $P_a$ and $P_b$ communicate $O(nm)$ encrypted elements. Hence, the total communication overhead is $O(\lambda nm)$ communication size in 2 communication rounds, where $\lambda$ is the computation security parameter employed in homomorphic encryption.

---

**Protocol 11: *RandomPair***

**Output:** Random pair $[\![\vec{r}]\!]$, $[\![\vec{r'}]\!]$, where $\vec{r'}[j] = \vec{r}[j]^2$.

1: $P_a$ randomly samples $[\![\vec{r}]\!]_a$, and $P_b$ randomly samples $[\![\vec{r}]\!]_b$.
2: **for** $j = 0$ to $m - 1$ in parallel **do**
3:     $P_a$ uses its private key to encrypt $[\![\vec{r}[j]]\!]_a$ to get $Enc([\![\vec{r}[j]]\!]_a)$, and sends $Enc([\![\vec{r}[j]]\!]_a)$ to $P_b$.
4:     $P_b$ randomly samples $t_j$ and computes $Enc(tmp_j) = 2 * Enc([\![\vec{r}[j]]\!]_a) * [\![\vec{r}[j]]\!]_b + t_j$.
5:     $P_b$ sends $Enc(tmp_j)$ to $P_a$.
6:     $P_a$ decrypts $Enc(tmp_j)$ to get $tmp_j$.
7:     $P_a$ computes $[\![\vec{r'}[j]]\!]_a = [\![\vec{r}[j]]\!]_a^2 + tmp_j$, and $P_b$ computes $[\![\vec{r'}[j]]\!]_b = [\![\vec{r}[j]]\!]_b^2 - t_j$.
8: **end for**

---

**Euclidean Triples vs. Random Pairs (Li et al., 2023).**: The communication overhead of *EuclideanTriples* is the same as the cost of generating ASS shares of random pairs in SecKNN (Li et al., 2023) based on homomorphic encryption. To securely compute the Euclidean square distance between two vectors $\vec{x}$ and $\vec{a}$, each of dimension $m$, SecKNN

requires generating ASS shares of a pair of random vector $\{\vec{r}, \vec{r'}\}$, each one of $m$ dimension. As is shown in Protocol 11, generating ASS shares of a random pair based on homomorphic encryption requires $O(\lambda m)$ communication size in 2 communication rounds. Hence, to generate ASS shares of $n$ random pairs to perform one KNN classification, SecKNN also requires $O(\lambda nm)$ communication size in 2 communication rounds.

