# OpenReview forum: "Kona: An Efficient Privacy-Preservation Framework for KNN Classification by Communication Optimization"
_ICML.cc/2025/Conference — ICML 2025 poster_

### Official Review · Reviewer_o6if · 2025-02-24

**Overall Recommendation:** 3

**Summary:**

This paper focuses on private KNN inference using multi-party computation techniques. Specifically, it introduces a new framework, Kona, for private KNN inference by designing new Euclidean triples. Additionally, the paper proposes a divide-and-conquer bubble protocol to achieve lower communication complexity. Finally, experiments are conducted to validate the effectiveness of the proposed method.

**Claims And Evidence:**

Yes.

**Essential References Not Discussed:**

No.

**Experimental Designs Or Analyses:**

It is recommended to enhance the experimental results by reporting the mean and standard deviation of repeated experiments in the tables. Conducting a statistical significance test, such as a t-test, to compare the proposed method with other methods would strengthen the empirical validation of the approach. Furthermore, the paper only compares the proposed method with one state-of-the-art method, which is insufficient.

**Methods And Evaluation Criteria:**

Yes.

**Other Comments Or Suggestions:**

There is a typo error in line 58 (left): The k in “k nearest neighbors’ selection” should be formatted as $k$.

**Other Strengths And Weaknesses:**

Strengths:

-	The paper addresses the problem of private KNN inference and proposes a new inference protocol by designing Euclidean triples. Furthermore, the paper optimizes the KNN inference framework by performing secure comparisons in parallel using a divide-and-conquer bubble protocol.
-	The paper conducts experiments on 8 datasets to validate the efficiency of the proposed method.


Weaknesses:

-	The contribution of the paper is somewhat limited. The divide-and-conquer bubble protocol appears to be a straightforward optimization, converting sequential comparisons into parallel comparisons. This protocol does not seem to have a direct relationship with the proposed Euclidean triple-based protocol, which raises questions about its novelty and integration.
-	As highlighted in the “Experimental Designs or Analyses”, the paper only compares the proposed method with one state-of-the-art method. This is insufficient to demonstrate the effectiveness and superiority of the proposed approach. Comparisons with multiple state-of-the-art methods are necessary to provide a more comprehensive evaluation.
-	The paper assumes that $k<<m<<n$. However, in the experiments, half of the datasets violate this assumption, as they exhibit $n<<m$. Additionally, for some datasets, $k\approx m$, which further deviates from the assumed conditions. This raises concerns about the practicality of the assumptions for real-world applications, as they may not hold in many scenarios.

**Questions For Authors:**

1.	Can the proposed method support private KNN inference with other distance measures?
2.	Intuitively, how does the proposed Euclidean triple help reduce the previously large communication overhead during private distance computation? Could you provide a high-level explanation of its mechanism?

**Relation To Broader Scientific Literature:**

This paper proposes new Euclidean triples for private KNN inference based on the input-independent but function-dependent technique [1].

[1] Ben-Efraim, Aner, Michael Nielsen, and Eran Omri. "Turbospeedz: Double your online SPDZ! Improving SPDZ using function dependent preprocessing." ACNS, 2019.

**Theoretical Claims:**

This paper does not have theoretical claims.

---

> ### Author Rebuttal · Authors · 2025-04-01
>
> Thank you for your insightful comments. Below, we respond to your concerns:
>
>
> ### 1. Can the proposed method support private KNN inference with other distance measures?
>
> Please see Point 1 for Reviewer F4RX.
>
>
>
> ### 2. Intuitively, how does the proposed Euclidean triple help reduce the previously large communication overhead during private distance computation? Could you provide a high-level explanation of its mechanism?
>
> At a high level, the Euclidean triple reduces the communication overhead by shifting the heavy communication of secure distance computation from the online phase to the offline phase. Traditional secure computation methods require interactive communication during the online phase of distance computation. This becomes a bottleneck when computing distances to many data points in real time. Our Euclidean triple addresses this issue by precomputing randomness for each distance computation before the actual query is known. Each Euclidean triple includes three parts: two random vectors and one precomputed squared distance between them. When the real query arrives, these random components allow the computation of the true Euclidean distance using only local operations. In other words, we replace communication with computation by embedding the necessary randomness and structure for secure distance computation into the Euclidean triples.
>
>
> ### 3. It is recommended to enhance the experimental results by reporting the mean and standard deviation of repeated experiments in the tables.
>
> Thank you for this valuable suggestion. In the final version, we will use a t-test to report both the mean and standard deviation of the results.
>
> ### 4. The paper only compares the proposed method with one state-of-the-art method, which is insufficient.
>
>  We initially focused on comparing our proposed Kona with the most recent and representative state-of-the-art (SecKNN, Li et al.~2023). During the rebuttal period, We further compare Kona with Liu et al.'s framework [1], which, aside from SecKNN, is the most recent state-of-the-art secure KNN framework and supports multiple data owners without leaking private information. The experimental results (detailed in https://gofile.io/d/TIx3Z2) show that Kona significantly outperforms Liu et al.'s framework by $24.62\times$ to $31.16\times$ in runtime. We will include the experimental results in the final version.
>
> [1] Liu et al. Toward highly secure yet efficient knn
> classification scheme on outsourced cloud data. IEEE
> Internet of Things Journal, 6(6):9841–9852, 2019.
>
> ### 5. The contribution of the paper is somewhat limited. The divide-and-conquer bubble protocol appears to be a straightforward optimization, converting sequential comparisons into parallel comparisons. This protocol does not seem to have a direct relationship with the proposed Euclidean triple-based protocol, which raises questions about its novelty and integration.
>
> We appreciate the reviewer’s perspective. Indeed, our proposed optimizations are complementary and independently valuable. Our proposed Euclidean triples focus on eliminating online communication during distance computation, addressing one major communication bottleneck. In parallel, our proposed divide-and-conquer bubble protocol targets the separate but equally critical bottleneck in secure neighbor selection by significantly reducing communication rounds through parallel comparisons and swaps. Although conceptually straightforward, this approach was previously unexplored in privacy-preserving KNN contexts. Previous studies either use an insecure method (discussed in Appendix D) or an inefficient sequential bubble protocol.   By combining these two optimizations, our proposed Kona achieves the first practical privacy-preserving KNN framework, which constitutes the novelty and practical contribution of our paper.
>
>
>
> ### 6. The paper assumes that $k \ll m \ll n$. However, in the experiments, half of the datasets violate this assumption, as they exhibit $n \ll m$. Additionally, for some datasets, $k \approx m$, which further deviates from the assumed conditions. This raises concerns about the practicality of the assumptions for real-world applications, as they may not hold in many scenarios.
>
>
> Thank you for highlighting this oversight. We acknowledge that our initial theoretical complexity analysis was presented under simplified assumptions ($k \ll m \ll n$). In the final version, we will carefully revise the theoretical complexity analysis to address more practical cases. Nevertheless, even without any assumption on the relationship among $k$, $n$ and $m$, the complexity of our proposed Kona remains favorable: the communication size is $O(kn)$ and the communication round is $O(k \log n)$. While the communication size and communication round of the baseline SecKNN is $O(mn + kn)$ and $O(kn)$. Thus, our conclusions about Kona's communication advantages remain valid even when there is no assumption on the relationship among $k$, $n$ and $m$.

---

> > ### Comment · Reviewer_o6if · 2025-04-05
> >
> > Thanks for the detailed response. The authors have addressed my concerns, and I would like to raise my score.

---

### Official Review · Reviewer_CiRr · 2025-03-12

**Overall Recommendation:** 3

**Summary:**

This paper presents a systematic framework called Kona for privacy-preserving KNN classification, which addresses communication inefficiencies in existing approaches. The key innovations include Euclidean triples to eliminate online communication for Euclidean Square Distance computations and a divide-and-conquer bubble protocol to significantly reduce communication rounds. The authors provide detailed security and communication complexity analyses, and the experimental results validate substantial efficiency improvements over  baseline methods across multiple datasets.

**Claims And Evidence:**

No

**Essential References Not Discussed:**

no

**Experimental Designs Or Analyses:**

The paper lacks clear discussion on how the number of attributes (dimension) affects computation time specifically. While Table 4 shows performance across datasets with different dimensions, a more explicit analysis of this relationship would strengthen the paper

**Methods And Evaluation Criteria:**

Most datasets contain a limited number of labels (10 or fewer), which may not fully demonstrate the framework's performance in scenarios with many class labels. Additionally, while the paper evaluates performance with k=5, analysis with different k values would provide more insight into the scalability of the approach.

**Other Comments Or Suggestions:**

I would think this paper is more like an security paper style, which is more systematical and with complete security analysis but may lack the novelty since KNN is not a popular algorithm recently. I would recommend the paper to be submitted to the top security conference.

**Other Strengths And Weaknesses:**

Strengths:
1. The paper presents a complete and systematic protocol for privacy-preserving KNN classification.
2. The security and communication analyses are thorough and well-executed.
3. The experimental results convincingly demonstrate Kona's efficiency improvements.

Weaknesses:
1. The protocol descriptions are complex and sometimes difficult to follow. A comprehensive pipeline diagram illustrating the data flow, encoding/decoding processes, and communications between parties would improve clarity.
2. While the paper mentions extensions to multi-party settings and vertically distributed data, these extensions lack detailed implementation or evaluation.

**Questions For Authors:**

1. How does the DQBubble protocol work under the encrypted distance, if all the encrypted distance use the same seed, will there be risk to leak seed such that disclose the original distance?

**Relation To Broader Scientific Literature:**

The paper adequately positions itself within the privacy-preserving machine learning literature and properly cites relevant work on secure KNN.

**Theoretical Claims:**

Yes, I have checked and did not find the fault.

---

> ### Author Rebuttal · Authors · 2025-04-01
>
> Thank you for your insightful comments. Below, we respond to your concerns:
>
> ### 1. How does the DQBubble protocol work under the encrypted distance, if all the encrypted distance use the same seed, will there be risk to leak seed such that disclose the original distance?
>
> Our proposed DQBubble protocol relies on secure comparison and multiplication protocols to `bubble' distances without leaking the original distance. Consider two additive secret-shared distances $[[x]]$ and $[[y]]$ with $x > y$. Our proposed DQBubble protocol first performs a secure comparison between $[[x]]$ and $[[y]]$ to obtain a secret-shared comparison result $[[comp]]$.  Then, the protocol securely swaps $[[x]]$ and $[[y]]$ based on $[[comp]]$ through:
> $ [[x']] = [[x]] * [[comp]] + [[y]] * (1 - [[comp]]), \quad [[y']] = [[x]] + [[y]] - [[x']]$. After this secure swap, although  $x'$ and $y'$ are equal to $y$ and $x$ respectively, $[[x']]$ and $[[y']]$ bear no recognizable relationship to $[[y]]$ and $[[x]]$ (e.g. $[[x']]_0$ and $[[x']]_1$ maybe 100 and -50, while $[[y]]_0$ and $[[y]]_1$ maybe 21 and 29). This is because the secure multiplication protocol introduces fresh randomness to mask $[[x']]$ and $[[y']]$.
>
> Regarding the use of seed, it is a common and secure practice in MPC protocols to use a single seed in a single execution.  A secure pseudorandom number generator ensures unpredictability for the seed and generated randomness, eliminating the risk of leaking seed and disclosure of original distances.  However,  reusing the same seed across multiple independent protocol executions should be insecure, because it would result in identical randomness sequences. Therefore, secure deployment always ensures that each protocol execution uses distinct seeds.
>
>
> ### 2. Most datasets contain a limited number of labels, which may not fully demonstrate the framework's performance in scenarios with many class labels.
>
> Our experimental datasets were selected primarily due to their wide use in KNN benchmarks, which typically do not encompass tasks with lots of labels. Moreover, the communication complexity of our proposed protocols is independent of the number of labels; thus, the efficiency advantages of our framework should consistently hold, irrespective of label number. In the final version, we will include datasets with more labels to validate this independence.
>
>
> ### 3. While the paper evaluates performance with k=5, analysis with different k values would provide more insight into the scalability of the approach.
>
> Please see Point 2 for Reviewer F4RX.
>
>
> ### 4. The paper lacks clear discussion on how the number of attributes affects computation time specifically.
>
> We will provide an analysis of how the number of attributes affects computation time in the final version.
>
>
> ### 5. The protocol descriptions are complex and sometimes difficult to follow. A comprehensive pipeline diagram illustrating the data flow, encoding/decoding processes, and communications between parties would improve clarity.
>
>
> We will add a comprehensive pipeline diagram to illustrate the data flow, encoding/decoding processes, and communications between parties in the final version.
>
> ### 6. While the paper mentions extensions to multi-party settings and vertically distributed data, these extensions lack detailed implementation or evaluation.
>
> Currently, our paper focuses on optimizing the communication overhead in the horizontal two-computation-party setting, which can support an arbitrary number of data owners and users and is foundational and widely applicable.  In the final version, we will elaborate on the necessary modifications for vertical data settings and multi-party settings.
>
> ### 7. I would think this paper is more like an security paper style, which is more systematical and with complete security analysis but may lack the novelty since KNN is not a popular algorithm recently.
>
> We believe our paper also holds significant novelty for premier AI conferences. Our paper systematically addresses the communication bottleneck of privacy-preserving KNN. Besides, our proposed optimizations—Euclidean triples and the divide-and-conquer bubble protocol—can benefit other privacy-preserving machine learning algorithms, such as K-Means and Approximate Nearest Neighbor (ANN), which also require Euclidean distance computation and neighbor selection. Additionally, KNN should be still a popular machine learning algorithm, particularly in medical diagnosis, finance, and natural language processing. A brief survey on DBLP identified more than 1,000 recent papers related to KNN (nearest neighbor) published within the past three years, including dozens appearing in premier AI conferences (e.g. ICML and NeurIPS) (references are available at https://gofile.io/d/TIx3Z2).

---

### Official Review · Reviewer_f4rX · 2025-03-18

**Overall Recommendation:** 3

**Summary:**

The paper introduces Kona, an efficient privacy-preserving framework for KNN classification, and optimizes communication overhead through two key methods: (1) It designs novel Euclidean triples to eliminate the need for online communication during secure computations of Euclidean squared distances. (2) It proposes a divide-and-conquer bubble protocol to reduce the number of communication rounds needed to select the k-nearest neighbors. Furthermore, the authors provide empirical results based on real data to support their findings.

**Claims And Evidence:**

The authors offered clear empirical analyses to support all of their claims.

**Essential References Not Discussed:**

The literature review is comprehensive and essential for understanding the paper's contributions.

**Experimental Designs Or Analyses:**

The experimental design effectively supports the claims regarding accuracy and efficiency. However, there is one issue that needs to be addressed: in Figure 2 of Section 4.4, which focuses on Ablation Evaluation, only the runtime, communication size, and number of communication runs are compared. The accuracy comparison is notably absent.

**Methods And Evaluation Criteria:**

1. The proposed methods make sense for real-world k-NN applications, particularly in the area of privacy preservation within the nearest neighbor community.

2. The evaluation criteria are appropriate for the problem. Both naive k-NN and state-of-the-art algorithms are presented and compared. The evaluations of accuracy and efficiency are relevant to the issue at hand.

**Other Comments Or Suggestions:**

1. Suggestion: Change the subscripts related to computation parts (e.g., P_0, P_1) to superscripts (P^0, P^1) to distinguish from users notation x_1, x_2.

**Other Strengths And Weaknesses:**

**Other Strengths:**

1. The motivation and structure of the paper are clear and easy for readers to understand.

2. The code is included in the supplementary materials, which is beneficial for practitioners.

**Other Strengths:**

1. The motivation and structure of the paper are clear and easy for readers to understand.

2. The code is included in the supplementary materials, which is beneficial for practitioners.

**Other Weaknesses:**

1. The proposed algorithm is limited to Euclidean distance for k-nearest neighbor (kNN) calculations, making it unusable with other popular distance metrics.

2. There is a lack of theoretical analysis and proof regarding the accuracy of the performance.

3. Some of the notations are not reader-friendly, and the subscripts can be confusing.

4. P3, right part, Share Comparison: [Z]=[x]<[y]. To be not ambiguous, it's better to add a parenthesis here: [Z]=([x]<[y])

5. In the experiment section, k=5 is small for kNN in a large size data. Generally, it should scale with the data size.

**Questions For Authors:**

1. Can the proposed methods be applied using distance metrics other than Euclidean distance?

2. Is it possible to conduct experiments with larger \( k \) values (e.g., 100) for a larger sample size to compare with the baseline?

3. Is there a method for selecting an appropriate \( k \) value based on different data sizes for the proposed algorithm?

**Relation To Broader Scientific Literature:**

This study provides comprehensive algorithm insights into privacy preservation issues for the k-NN method. The communication size and number of communication rounds have seen significant improvements compared to existing methods. This is beneficial for the nearest neighbor community.

**Theoretical Claims:**

This is not relevant to theoretical claims. The paper concentrated on designing algorithms.

---

> ### Author Rebuttal · Authors · 2025-04-01
>
> Thank you for your insightful comments. Below, we respond to your concerns:
>
>
>
> ### 1. Can the proposed methods be applied using distance metrics other than Euclidean distance?
>
> Yes, both of our proposed optimizations can be applied to other distance metrics.
>
> First, our proposed divide-and-conquer bubble protocol makes no assumption on the distance metrics. Thus, it can be applied to any distance metrics.
>
> Besides, our proposed Euclidean triple optimization can be applied to other distance metrics with minor adaptations. That is because the core idea of our optimization is to use the input-independent but function-depend technique (correlated randomness) to reduce online communication, and this idea can apply to any distance metrics. For example, for cosine similarity ( $\frac{\vec{x} \cdot \vec{a}}{\|\vec{x}\|\|\vec{a}\|}$), whose main communication bottleneck lies in secure dot product computations ($\vec{x} \cdot \vec{a}$, $\vec{x} \cdot \vec{x}$ in $\|\vec{x}\|$, and $\vec{a} \cdot \vec{a}$ in $\|\vec{a}\|$), we can adapt our Euclidean triple to be a dot triple, which comprises two secret-shared vectors and their dot products.  By leveraging the dot triple, all secure dot product computations in cosine similarity can be executed without online communication. For Hamming distance, which is usually used in binary data, the Euclidean triple can directly apply in the binary domain because $(x - a)^2$ equals the mismatch indicator. Thus, the sum of squared differences immediately gives the Hamming distance for binary vectors.
>
> We will discuss how our proposed methods can be applied to other distance metrics in detail in the final version.
>
> ### 2. Is it possible to conduct experiments with larger ( k ) values (e.g., 100) for a larger sample size to compare with the baseline?
>
>
> Yes, conducting experiments with larger values of $k$ is feasible. Our proposed framework, Kona, places no constraints on the value of $k$. Besides, the benefits of our optimizations become even more pronounced with increasing $k$. That is because the communication complexity in Kona is $O(k\log n)$, while the communication complexity in  SecKNN is $O(kn)$. The performance gap widens significantly for larger $k$. For instance, preliminary experiments using $k = 100$ on the Mnist dataset yield substantial reductions: the runtime in the WAN setting decreases from 248124 seconds (about $68$ hours) to $1084.35$ seconds (a $265\times$ reduction), the communication rounds decrease from $13989901$ (SecKNN) to $3417$ (Kona) (a $4094.20\times$ reduction), and the communication size decreases from $942.91$ MB to $503.87$ MB (a $1.87\times$ reduction).   We will include an evaluation of different $k$ in the final version to showcase Kona's scalability better.
>
> ### 3. Is there a method for selecting an appropriate ($k$) value based on different data sizes for the proposed algorithm?
>
> Yes, we can select an appropriate $k$ value based on different data sizes for our proposed algorithm, because our proposed algorithm directly accepts $k$ as an input parameter. We can employ standard selection methods, such as the method proposed by Maleki et al.[1], to select an optimal $k$, and then input the $k$ to our proposed algorithm.
>
>
> [1] Maleki et al. A novel simple method to select optimal k in k-nearest neighbor classifier[J]. International Journal of Computer Science and Information Security, 2017, 15(2): 464.
>
>
> ###  4. However, there is one issue that needs to be addressed: in Figure 2 of Section 4.4, which focuses on Ablation Evaluation, only the runtime, communication size, and number of communication runs are compared. The accuracy comparison is notably absent.
>
> We omitted accuracy comparisons in the ablation evaluation because both of our optimizations exclusively target communication efficiency, without affecting the classification accuracy. As is shown in Table 3 of our paper, Kona consistently achieves identical accuracy to scikit-learn across all datasets. This result demonstrates that our optimizations do not affect the classification accuracy. For completeness, we will include an accuracy comparison to the ablation evaluation in the final version.
>
>
> ###  5. There is a lack of theoretical analysis and proof regarding the accuracy of the performance.
>
> In the final version, we will include a theoretical analysis to clearly explain why the accuracy of our proposed protocols is equivalence with plaintext KNN classification. A brief explanation is that our secure KNN classification protocols fundamentally replicate the plaintext KNN classification exactly, and thus theoretically preserve identical accuracy.
>
> ### 6. Some of the notations are not reader-friendly, and the subscripts can be confusing.
>
> We will revise the notations to make them more reader-friendly in the final version.

---

> > ### Comment · Reviewer_f4rX · 2025-04-02
> >
> > Thank you for your detailed rebuttal to my reviews. All my concerns have been addressed. I will maintain my overall ratings and I'm pleased to see that some of my suggestions may appear in the final version.

---

### Decision · Program_Chairs · 2025-05-01

**Decision:**

Accept (poster)

**Comment:**

This paper presents a communication-efficient privacy-preserving kNN classification framework based on (i) a novel Euclidean triples that enable secure Euclidean square distance computations without any online communication and (ii) a divide-and-conquer bubble protocol that reduces the communication rounds for selecting neighbors. Experimental results demonstrate superior efficiency of the proposed framework as compared with the existing framework.

---

The paper is well-written and addresses a practical problem with significant improvements. A noted weakness is its lack of technical depth from a traditional machine learning perspective, but I think ICML has an open mind to diverse tastes and hence this should not undermine the value of the work.

On the technical side, the proposed method seems reasonable and effective. Reviewer o6if points out a mismatch between the assumptions for complexity analysis (k<m) and experiment setting (k~m) and the lack of experimental comparisons with existing methods. Authors made a plausible effort of responding in rebuttal, and the reviewer raised score afterwards.

---

Overall, I think the paper makes an acceptable contribution and could be accepted if there is room in the program. Authors are also encouraged to incorporate all reviews in the final version, especially align the assumptions and clarify the issues raised in the reviews.